# Properties of Rubberized Concrete Prepared from Different Cement Types

**Lamiaa K. Idriss** [1,]*[⬛] **and Yasser Abdal Shafey Gamal** [2]

1   Department of Civil Engineering, Sphinx University, Assiut 71515, Egypt
2   Department of Civil Engineering, High Institute of Engineering Technology, EL-Minia 61511, Egypt;
    yasser.gamal2310@mhiet.edu.eg
*   Correspondence: lamiaa.idriss@sphinx.edu.eg

**Abstract:** At present, global waste tire generation considerably exceeds consumption. Moreover, waste rubber tires (WRTs) are a cause of concern, as huge volumes are being discarded and buried, thus causing serious environmental pollution. Rubberized waste concrete (RWC) is a type of environmentally friendly construction material. The main challenge encountered when manufacturing rubberized concrete is the low adhesive properties between the cement paste and rubber particles. This paper demonstrates the effects, through experiments, of using waste tire rubber instead of recycled coarse aggregate (RCA) on two types of cement, i.e., sulfate-resistant cement (SRC) and ordinary Portland cement (OPC), where SRC is a specially blended cement designed to improve concrete performance and workability in the most aggressive environments. All tested samples contained 10% silica fume (SF) and 0.2% fly ash (FA), and the substitution of recycled aggregate content with waste rubber tier (WRT) at different percentages of 100%, 75%, and 50% was evaluated. The research investigated the synergistic effect on the workability and mechanical properties of various cement types with different amounts of rubber aggregate. It was found that the sulfate-resistant (SRC) type can increase the compressive strength than OPC with a percentage of 25% with the same content of WRT at concrete mix. Moreover, ductility and cracking behavior are improved, and it appears that it is also possible to make lightweight rubber aggregate concrete with this type of mixture. Using this type of cement, it is possible to restore satisfactory ductility to the waste tires, thus facilitating a reduction in the formation of potential plastic cracks. Moreover, the indicative compressive strength development for SRC with recycled rubber in concrete positively contributes to a reduction in formed cracks. However, SEM microstructural analyses suggest a higher proportion of C–S–H intermixed with sulfate reaction phases of SRC rubberized mortar than those of OPC; thus, given that crystal growth results in a decreased percentage of air voids rather than decreased internal cracking, it is clearly shown that the average crack width increases in OPC mortar compared with SRC. Finally, *t*-testing was used as an inferential statistical tool to determine whether there is a sizeable distinction between the properties of the two categories of materials, OPC and SRC, by comparing the mean and standard deviation of the values for compressive and tensile strength.

**Keywords:** waste rubber tires; recycled aggregate; cement mortar; silica fume; ductility; cracking performance

## 1. Introduction

Solid waste management is a serious environmental issue for towns all over the world. Waste tires are a major source of concern because they are not biodegradable, resulting in the majority of waste tires accumulating; waste tire disposal has consequently become one of the most critical environmental issues in recent years. Due to the growing depletion of existing waste disposal sites, waste tire disposal in landfills may become unfeasible in the future [1]. Such dumps pose a health and environmental risk to the communities in which they are located. At this time, it is projected that roughly one billion tires are withdrawn

from users around the world every year [2]. Over half of the waste tires are disposed of without being treated, making waste tire disposal a major environmental issue in cities around the world. The simplest and cheapest method for tire disposal is to burn them, but this causes pollution, which is unpleasant and even illegal in many countries.

By comparison, the use of waste tires in the fabrication of concrete may have the following advantages regarding concrete properties: (1) reduced modulus of elasticity, which also depends on the content of recycled rubber in the concrete; (2) increased toughness and resistance to vibration of the concrete; and (3) increased abrasion resistance of concrete. Thus, the foundation of this strategy is the potential for cementitious aggregates to be improved in terms of technological engineering criteria (thermal–acoustic insulation, energy dissipation capacity, and durability), while also being environmentally friendly.

The incorporation of waste tires into cement-based materials is a viable approach for industrial waste reuse and recycling. Furthermore, this strategy may result in a significant reduction in the consumption of natural resources (sand, water, and coarse mineral aggregates) required for the manufacture of building materials [3].

Gayathri and Raja [4] investigated the mechanical characteristics of concrete produced using rubber crumb and microsilica, where microsilica was used as an admixture (Portland cement substitute) at percentages of 5%, 10%, and 15%, and rubber crumb was also added at 5% from fine aggregate. Accordingly, the findings demonstrate that the use of recycled rubber causes a 15% decrease in the concrete tensile strength and, moreover, mixing of both recycled rubber and microsilica at 10% each results in increasing the flexural strength.

Habib [5] investigated the mechanical and dynamic characteristics of high-strength concrete made from both coarse and fine rubber crumb tires, where steel fibers and microsilica were also used, and the rubber crumb was also added to substitute fine and coarse aggregates at proportions of either 15% or 25%.

The cracking behavior of concrete is a critical factor in its failure mechanism. Considering another perspective, concrete cracking behavior can be altered by using light and recyclable elements such as crumb rubber. Gupta examined the fracture behavior of rubberized concrete beams with different water to cement ratios ranging from 0.35 to 0.55, and the fracture parameters were determined using fracture and size effect methods [6]. Ghewa GJP and Suprobo P discovered that brittleness number, fracture toughness, and total fracture energy increased, while both the effective crack tip slot displacement at maximum load and fractures decreased [7].

According to laboratory testing, incorporating discarded tire rubbers into the design of a concrete mixture improves impact resistance, sturdiness, and plastic deformation properties, indicating that the concrete has substantial potential for application in retaining structures, sound/crash barriers, and pavements. Despite the fact that almost all previous studies have found a significant reduction in concrete strength, there are various advantages to using tires in concrete, including their light weight, enhanced thermal insulation, high permeability, and vibration absorption [8]. The substitution of natural aggregate with rubber crumb reduces the compressive and flexural strengths of high-strength concrete, and, therefore, rubber crumb has experienced increased attention for its application in the design of high-strength concrete [9]. Rubber crumb from tires was used by Thomas and Gupta [10] in various concrete formulations, where it was used to substitute fine aggregate in the range of 0% to 20%. Experimental formulations were also prepared using water coefficients of w = 0.4, 0.45, and 0.50. P Martauz and V Vaclavik [4] found that using rubber crumb from tires with fibers increased the toughness index of concrete, and that the use of recycled rubber tires improves impact energy absorption.

P Martauz and V Vaclavik concluded that the concrete made from recycled rubber has a stronger abrasion and wear resistance. Moreover, as the percentage of natural aggregate replaced by rubber crumb in concrete increases, the thermal conductivity of the concrete drops. However, the concrete thermal conductivity coefficient was 0.27–0.34 when 100% natural aggregate was replaced with rubber crumb. In addition, the concrete made with crushed rubber and fibers had better physical and mechanical characteristics than those

of pure rubber crumb concrete [4]. Samuel and Seckley [11] investigated the mechanical strength of concrete made from recycled tire rubber, whereby recycled rubber was used to replace natural aggregate in the following proportions: 2.5%, 5%, 7.5%, 10%, 12.5%, 15%, 17.5%, and 20%. It was deduced that the addition of 2.5% rubber crumb increased compressive strength by around 8.5%.

Considering the use of waste tire rubber, combined with existing WRT uses in the construction sector, it may be possible to utilize all waste tires and alleviate the problems connected with their disposal. Moreover, these uses can contribute to eliminating the issue of tire disposal, specifically in Egypt. Therefore, this research focuses on the engineering features of tire-rubber-modified concrete and highlights achievements due to the inclusion of tire rubber material in mixtures. In addition, methodologies for enhancing the properties and characteristics of the rubberized concrete are presented. Furthermore, density, porosity and void ratio, slump, compaction factor, and water absorption were assessed in fresh concrete tests, and the tensile strength and compressive strength were also measured after 7, 14, and 28 days.

The study focused on experimentally analyzing the effect of cement type on the properties of rubberized concrete made with different amounts of WRT replacing RCA as coarse aggregate. The two types of cement mortars were SRC and OPC, and the characteristics of the final rubberized concrete were evaluated as a basis for determining the optimal concrete mix. All mixtures contained 10% SF and 0.2% FA, and the percentages of WRT used to replace recycled aggregate content were 0%, 50%, 75%, and 100%.

## 2. Materials

### 2.1. The Procedures of Experiment

The categorization of classification was created according to the type of cement, with the first series corresponding to OPC, having the sample codes of RF, RO1, RO2, and RO3. The second series corresponds to SRC, and the codes are RS4, RS5, and RS6. The water content to cement ratio ($w/c$) was kept constant at a value of 0.55 for all admixtures [5]. Furthermore, different experiments were set up with the aim of enhancing the behavior of the rubberized concrete. Slump loss, compaction factor, water absorption, contained air, and bulk density were determined in fresh concrete tests, whereby nine cubes from mold specimens having dimensions of 150 mm × 150 mm × 150 mm were used for each sample, and the compressive strength was tested at 7, 14, and 28 days. Additionally, six cylinders of 150 mm diameter and 300 mm length were used for split tensile strength testing at 14 days, and we also studied ductility performance and the failure mechanism. Finally, scanning electron microscopy (SEM) was used to study the interaction between the components for both OPC and SRC; this was performed using a Scanning Electron Microscope Model (JSM-5400 LV) with a Quantitative and Qualitative Analysis Unit for Minerals in the Sample (WDS + EDS)—Japan EDAX Japan KK 13-31, Kohnen 2-chome Minato-ku, Tokyo 108-0075 Japan.

### 2.2. Aggregate

WRTs were used as coarse aggregate, in addition to cuttings of crumb rubber that had been produced from automotive and truck scrap tires in Assiut City. The waste tires were cut into small pieces without separation between steel and textile to reduce the recycling process and hence save costs through reuse; the size of crumb rubber ranged from 150 to 450 mm, and the bulk density was 110 kg/m³. The tires also comprised rubber, metal, sulfur, textile, carbon black, zinc oxide, and other additives. The RCA used in the experiments had a maximum aggregate size of 20 mm.

Figure 1 demonstrates the steps of recycling the waste tires and the recycled coarse aggregate, and Table 1 details the properties of RCA, and the fine and coarse sand, used in this study. Finally, Figure 2 presents the passing percentage curve for all aggregates and waste tires that were used in the testing.

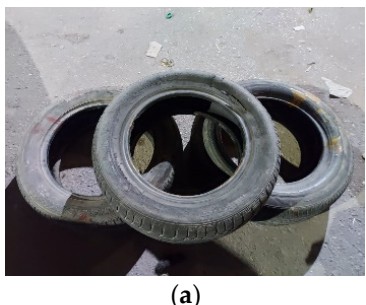 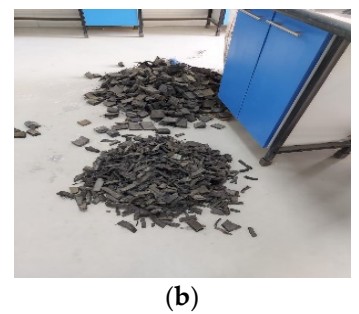 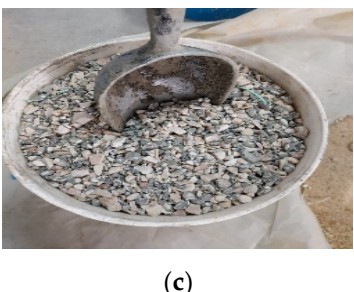

(**a**)        (**b**)        (**c**)

**Figure 1.** Recycled waste tire and coarse aggregate: (**a**) waste tires; (**b**) rubber particles from the shredding process; (**c**) recycled coarse aggregate.

**Table 1.** Coarse and fine aggregate properties.

| Properties | Values for WRT | Values for RCA | Values for Sand |
|---|---|---|---|
| Specific gravity | 1.05 | 2.65 | 2.53 |
| Density gm/cm$^3$ | 1.10 | 1.25 | 2.92 gm/cm$^3$ |
| Water absorption (%) | 1.05 | 3.06 | 2.6 |
| Void ratio | - | 0.525 | 0.154 |
| Crushing value (%) | 10 | 21.80 | - |
| Total porosity | - | 0.20 | 0.133 |
| Fineness modulus | - | 7.11 | 2.30 |

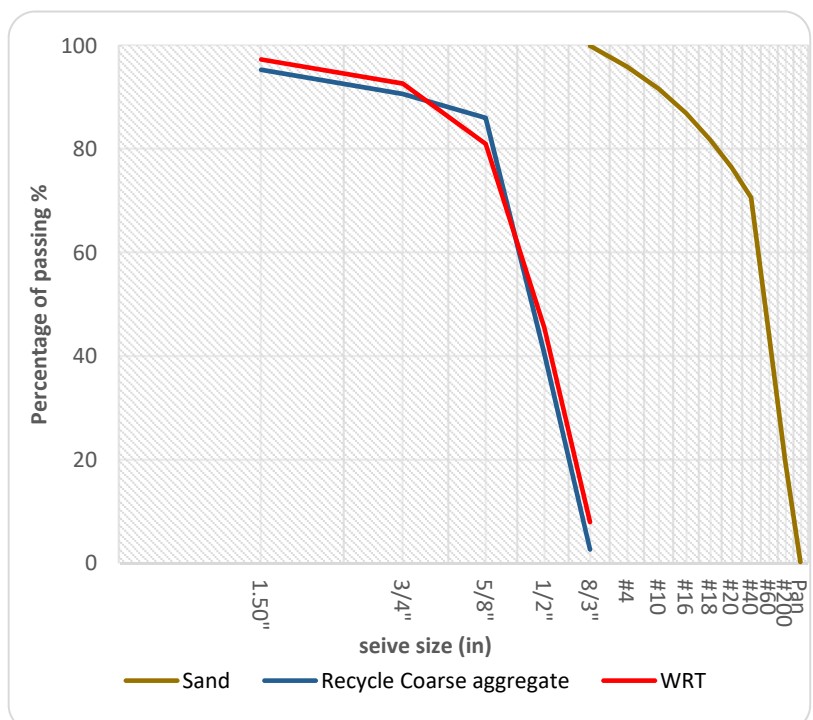

**Figure 2.** Sieve analysis curves for sand, RCA, and WRT.

### 2.3. Cement

Two types of cement were used in the mixtures. The first, SRC (CEM IV/A (P) 42.5 N–SR), has very high compressive strength and high durability under aggressive conditions, and protects structures from sulfate attack. Furthermore, its very low heat of hydration helps to avoid shrinkage cracks. The second was OPC(CEM II/B-P (42.5 N)). Both cements

were tested according to ESS (4756/1/2013) [12] and BS EN (197-1/2011) [13] standards. SF and FA were used as additive materials, and it is well established that the use of suitable FA results in the improvement of most of the properties of concrete. In addition to economic and ecological considerations, the use of fly ash in concrete also results in improvements regarding workability, bleeding, reduced separation, heat evolution, and permeability, along with improved ultimate strength, enhanced sulfate resistance, and inhibition of alkali aggregates. SF was added as cementation material to the concrete mix, and is able to augment the mechanical properties and achieve better corrosive strength, increased sulfate resistance, and superior abrasion resistance. In addition, using silica fume in concrete helps to reduce the permeability of the mixture and improve its durability. A superplasticizer referred to as Sikament (hereafter Sikament-163 M), which conforms to ASTM C 494-92 Type F, was also added initially to accelerate hardening and increase ultimate strength. All properties of each cement and superplasticizer are presented in Tables 2 and 3. Table 4 shows the composition of Sikament-163 M. FA, SF, and Sikament were produced by the Sika company in Egypt. The effect of the prepared superplasticizers was assessed by comparing the admixture dosages' effect on the combined water content and the compressive strength; it was evident that, as the admixture dosages increase, the combined water content decreases and the strength improvement is enhanced [14,15].

**Table 2.** The properties of cement types.

| Type of Cement | Specific Gravity kg/m$^3$ | % of Water Consistency | Specific Surface Area (Bline) m$^2$/kg | Bulk Density (kg/m$^3$) | Soundness (Le Chatelier) m |
|---|---|---|---|---|---|
| Ordinary Portland Cement | 3079.288 | 28 | 282.7 | 1140 | 0.00015 |
| Cement Sulfate Resisting | 3079.288 | 28 | 406.6 | 1140 | 0.00015 |
| Type of cement | Initial setting-time (min) | Final setting time (min) | 7 Days | Compressive strength (N/m$^2$) 14 Days | 28 Days |
| Ordinary Portland Cement | 180 | 230 | $3.8 \times 10^7$ | $5.2 \times 10^7$ | |
| Cement Sulfate Resisting | 145 | 195 | $3.7 \times 10^7$ | $5.0 \times 10^7$ | |

**Table 3.** The properties of silica fume (SF) and Fly Ash(FA) from the manufacturer data sheet.

| Silica Fume (SF) | | | | | |
|---|---|---|---|---|---|
| Type of Additives | Specific Gravity kg/m$^3$ | Mean Grain Size | Specific Surface Area (Bline) m$^2$/kg | Bulk Density (kg/m$^3$) | Silica Fume Particles |
| SF | 2200 | 7 | 17,800 | 4300 | 0.50 µm |
| Fly Ash(FA) | | | | | |
| Type of Additives | Specific Gravity kg/m$^3$ | SiO$_2$ | Sulphate SO$_4$ | Al$_2$O$_3$ | CaO | MgO |
| FA | 2120 | 56.88% | 0.27% | 27.65% | 3.6% | 0.34% |



**Table 4.** Properties of superplasticizer Sikament 163 M(SP) from the manufacturer data sheet.

| Properties | Sikament (SP) |
|---|---|
| pH value | 4.3–4.7 |
| Density at 20 °C | 1.20 kg/L |
| Fineness modulus | 3.06 |
| Void ratio | 0.52 |

### 2.4. Preparation of the Specimens

A drum mixer was used to mechanically mix the components of the concrete mixtures, with WRT with RCA and fine aggregate (sand) mixed for thirty seconds followed by the addition of both of silica fume (SF) and water. Sikament 163 was added to one liter of water with an amount of 0.55% (based on cement mass) to form an aqueous solution. In total, seven concrete mixtures were tested with different types of cement, i.e., OPC and SRC. In addition, all mixtures were designed with a constant water/binder ($w/b$) ratio of 0.55 and binder content of 396 kg/m$^3$. Rubberized concrete mixtures were prepared with SF and FA, by percentage 10% and 0.2% from cement weight respectively according to [3]. Mixtures were designed for all combinations, and the percentages of WRT used to replace recycled aggregate content were 100%, 75%, and 50%, as shown in Table 5. The fresh and hardened concrete was designed according to ECP 203-2007 (2007) Egyptian Code [13,16]. A slump test was conducted and the compact factor and bulk density were investigated to determine the workability of the concrete mix. A total of 54 cubic molds having dimensions of 15 mm × 15 mm × 150 mm were made for determination of compressive strength at 7, 14, and 28 days. Cylindrical molds having dimensions of 150 mm × 300 mm were used for splitting tensile tests at 14 days, where all specimens were covered with a plastic sheet and kept in molds at room temperature for 24 h. All specimens were demolded and immersed in a water tank for 28 days at the temperature of the laboratory (23 ± 5 °C); the steps of the mixing process of samples are shown in Figure 3.

**Table 5.** Mix proportions for rubberized concrete tests.

| Mix No. | Cement (kg/m$^3$) | | | | Fine Aggregate (Sand) (kg/m$^3$) | | Coarse Aggregate (kg/ m$^3$) | | w/c |
|---|---|---|---|---|---|---|---|---|---|
| | Type | Weight | FA | SF | SP | Type | Weight | RCA | WRT | |
| RF | OPC | 396 | 0 | 0 | 0 | | 804 | 846 | 0 | 0.55 |
| RO1 | | 396 | 0.795 | 39.6 | 12 | | 804 | 0 | 846 | 0.55 |
| RO2 | OPC | 396 | 0.795 | 39.6 | 12 | Siliceous sand | 804 | 212 | 634 | 0.55 |
| RO3 | | 396 | 0.795 | 39.6 | 12 | | 804 | 423 | 423 | 0.55 |
| RS4 | | 396 | 0.795 | 39.6 | 12 | | 804 | 0 | 846 | 0.55 |
| RS5 | SRC | 396 | 0.795 | 39.6 | 12 | | 804 | 212 | 634 | 0.55 |
| RS6 | | 396 | 0.795 | 39.6 | 12 | | 804 | 423 | 423 | 0.55 |

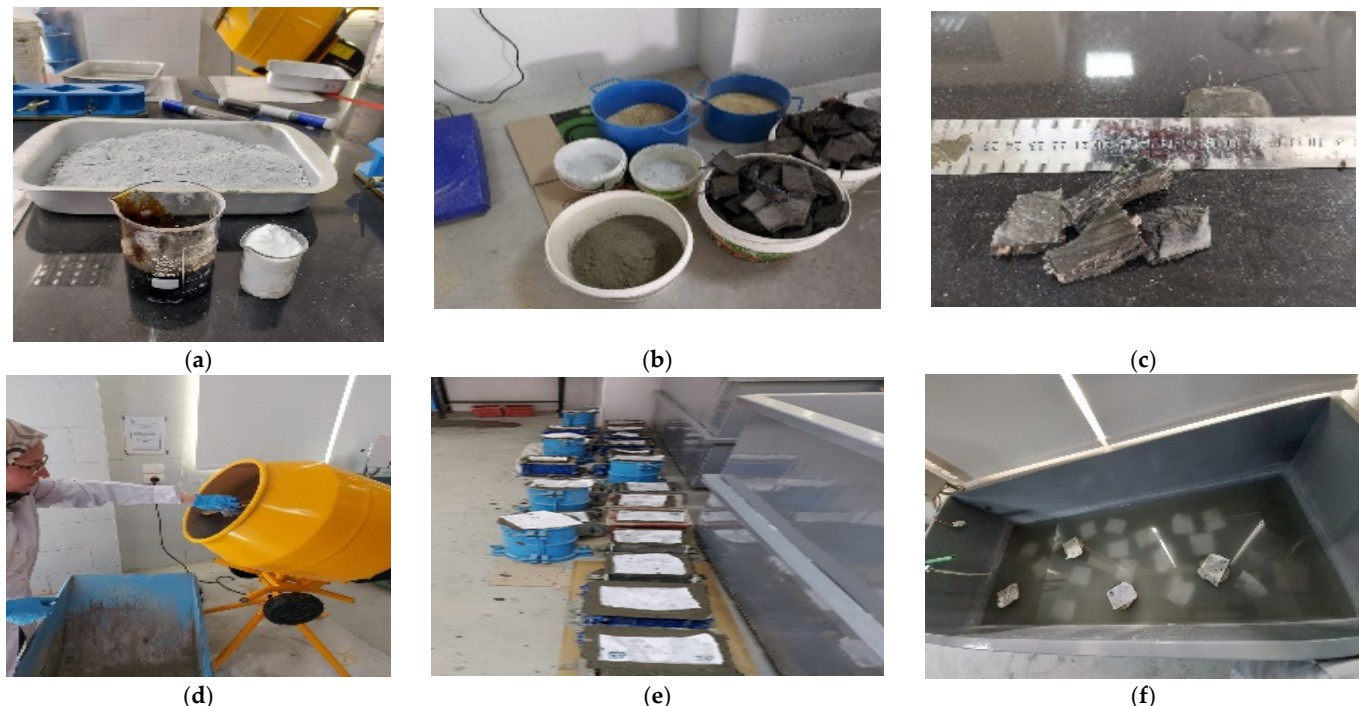

**Figure 3.** Mixing and preparation of specimens: (**a**) Sikament, FA, and SF; (**b**) cement, sand, and waste tires; (**c**) size of WRT; (**d**) a pan mixer; (**e**) cubic and cylinder specimens; (**f**) water tank curing.

## 3. Results and Discussion

### 3.1. Workability

Slump and compacting factor tests are the most important tests for measuring the workability of concrete. The tests were conducted according to ES: 1658-2/2008 and ECP 203-2007 [12,13,16], as shown in Figure 4, and the degree of concrete workability was studied in rubberized concrete for two different types of cement, OPC and SRC. Figure 5a shows the increase in the slumping values. These findings show that SRC has the highest workability, even though there was a loss in slump of 150–130 mm; in contrast, this value for the concrete mix design with OPC was 130–120 mm, and varied according to the percentage with respect to ordinary concrete, RF. Compared with the reference sample RF, the slump values for OPC samples (RO1, RO2, and RO3) with percentages of 100%, 75%, and 50% WRT were 243%, 271%, and 243% higher, respectively. In addition, the slump test values for SRC samples RS4, RS5, and RS6, with the same series of WRT percentages, were higher than those of the RF by 243%, 329%, and 271%, respectively. Furthermore, the increase in slump for different percentages of WRT samples leads to increased workability. Thus, the results are consistent with previous studies, which showed good agreement with workability measurements, with the use of WRT achieving higher workability compared with the ordinary concrete control, as studied by Raghavan and Huynh [17,18]. Table 6 shows the values of slump and compaction factors for all studied samples of OPC and SRC. The slump increases with the inclusion of WRT for both OPC and SRC because water absorption increases. Hence, the initial porosity increases and the density decreases compared to the reference RF sample.

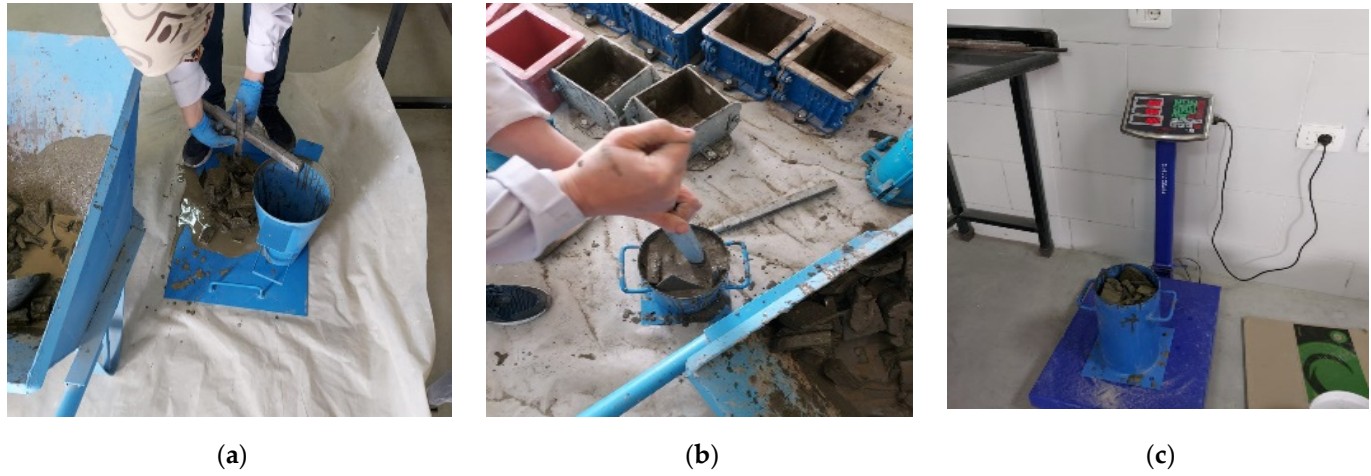

**Figure 4.** Fresh concrete tests: (**a**) slump test; (**b**) compact factor; (**c**) bulk density.

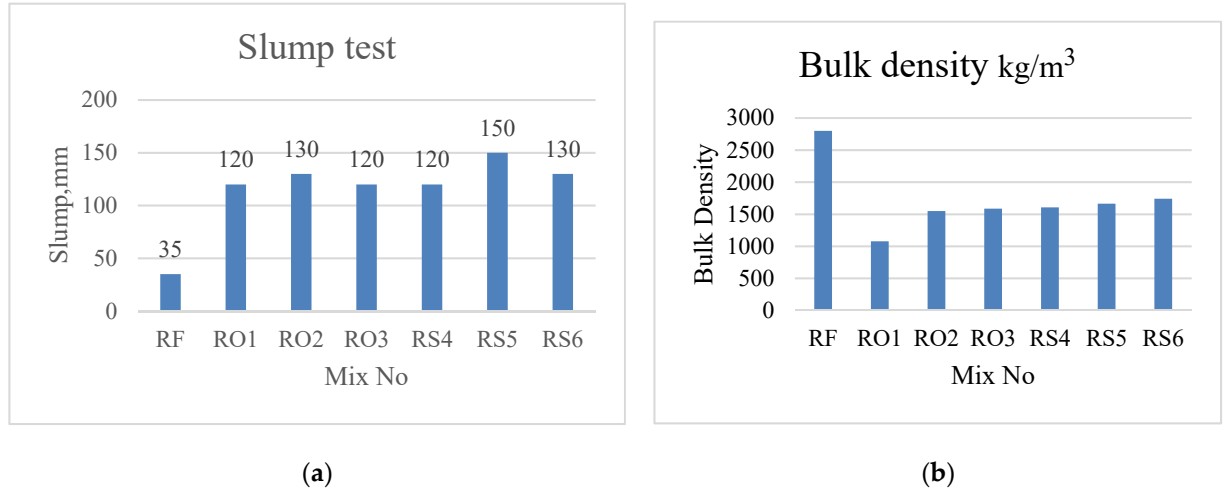

**Figure 5.** Fresh WRT concrete tests: (**a**) slump test; (**b**) bulk density.

**Table 6.** Fresh concrete test results.

| Mix. No. | Slump mm | Compact Factor | Bulk Density kg/m$^3$ | Workability |
|---|---|---|---|---|
| RF | 35 | 0.93 | 2800 | low |
| RO1 | 120 | 0.97 | 1077 | High |
| RO2 | 130 | 0.97 | 1550 | High |
| RO3 | 120 | 0.97 | 1587 | High |
| RS4 | 120 | 1.0 | 1606 | High |
| RS5 | 150 | 1.0 | 1663 | High |
| RS6 | 130 | 1.0 | 1740 | High |

*3.2. Bulk Density*

The bulk density of concrete specimens with varying WRT ratios was evaluated to determine the effect of cement type by employing WRT as a partial replacement for RCA in terms of total weight, as indicated in Table 6. Cement types were taken into account as another factor to study their impact on results, and Figure 5b shows the changes in the densities according to sample components. The results demonstrate that the bulk densities

for OPC samples (RO1, RO2, and RO3) having WRT percentages of 100%, 75%, and 50% are lower than those of the reference sample (RF) by 62%, 45%, and 43%, respectively. In addition, the values for SRC samples (RS4, RS5, and RS6) with the same series of WRT percentages are smaller than those of the RF by 43%, 41%, and 38%, respectively. The densities are higher for SRC than for OPC. Therefore, the type of cement has an effect on increasing the densities, and increasing the WRT content also has an impact on reducing density; the density of RO3 (50% WRT) is higher than that for RO1 by 47%, and that of RS6 (50% WRT) is also higher than that for RS4 by 8%. The density values of rubberized concrete for the two cement types match with the range of densities in lightweight concrete, where LWC has values between 320 to 1920 kg/m$^3$ according to ACI 213 [19]. Hence, rubberized concrete can be used in the construction of buildings as LWC.

### 3.3. Water Absorption, Porosity, and Air Content

For both OPC and SRC samples, water absorption, porosity, and air content tests were used to investigate the impact of WRT and the type of cement, as shown in Table 7 and Figure 6. However, adding SF to all samples contributes to reducing the water absorption [20–22]. Table 7 shows the water absorption ratio (WAR) for all samples according to ASTM C642-81 [23]. Figure 6a shows the dramatic decrease in WAR for OPC samples with reductions in the amount of WRT; however, for SRC samples, there is a slight decrease when decreasing the amount of WRT from 100% to 50%. From the results of the first series (OPC), it is clear that the water absorption ratios for OPC samples (RO1, RO2, and RO3) with WRT percentages of 100%, 75%, and 50% are lower than those of the reference sample (RF) by 306%, 197%, and 80%, respectively. In addition, the values for SRC samples (RS4, RS5, and RS6) with the same series of WRT percentages are lower than those of the RF by 181%, 166%, and 158%, respectively. That is, the water absorption ratio increases by substituting WRT content for RCA to 100%, which is due to the weak bond between cement paste and rubber aggregate. Moreover, the WAR for samples with OPC undergoes a significant change in value due to reducing the WRT percentage (from 100% to 50%), and the WAR for SRC samples is slightly reduced with a reduction in the amount of WRT. The porosity tests involve several stages according to ACI 318-19: Building Code Requirements and Commentary. First, the samples are oven dried for 2 h. at 105 °C before being weighed, and second, the dry samples are soaked in water for 24 hr., after being placed in a water tank saturation machine and evacuated for 2 hr. The damp specimen surfaces are then wiped with a cloth before the samples are weighed. The porosity values are determined as the ratio of the weight difference before and after soaking in water divided by the concrete volume. Different samples were tested for this, including the RF for comparison. Figure 6b shows the values of measurements for different cement types. The results demonstrate that, for the first series (OPC), the porosity ratios for OPC samples (RO1, RO2, and RO3) with WRT percentages of 100%, 75%, and 50% have values higher than those of the reference sample (RF) by 246%, 168%, and 54%, respectively. In addition, the values for SRC samples (RS4, RS5, and RS6) with the same series of WRT percentages are higher than those of the RF by 179%, 73%, and 152%, respectively. The increasing air content of samples is associated with low carbonation and chloride durability [24], and this affects the concrete characteristics. Therefore, the air content was investigated to analyze the impact of different types of cement on WRT. The void ratios for OPC samples (RO1, RO2, and RO3) have values lower than those for the reference sample (RF) by 19%, 22%, and 72%, respectively. In addition, the values for SRC samples (RS4, RS5, and RS6) are smaller than those of the RF by 28%, 35%, and 54%, respectively, as shown in Figure 4c; that is, the void ratios increase with increasing amounts of WRT. In addition, the void ratios are higher for RO1 and RO2 than for RS4 and RS5. Therefore, OPC samples with WRT of 100% and 75% have higher values compared with the corresponding SRC samples. However, in the case of 50% WRT, the air content is higher for SRC (RS6) than OPC (RO3); that is, there is a significant decrease in the void ratios for both OPC and SRC mixtures. Therefore, it can be seen that the disadvantage of lowering the air content is that it may lead to a reduction in some of

the desired effects of using tire rubber in concrete. This means that air-entrained concrete will have a low content of air voids. Optimal air entrainment will protect concrete from the harmful effects of sulfate attack and freeze–thaw damage.

**Table 7.** Mechanical properties of samples after 7, 14, and 28 days.

| Mix. No. | Compressive Strength (MPa) | | | Tensile Strength (MPa) | Water Absorption (MPa) | Voids Ratio | Porosity Ratio |
|---|---|---|---|---|---|---|---|
| | 7 Days | 14 Days | 28 Days | 14 Days | 14 Days | 14 Days | 14 Days |
| RF | 4.62 | 7.10 | 14.10 | 1.10 | 1.28 | 5.60 | 1.88 |
| RO1 | 0.88 | 1.0 | 2.20 | 0.45 | 5.20 | 4.56 | 6.50 |
| RO2 | 2.30 | 2.40 | 4.20 | 0.65 | 3.8 | 4.36 | 5.03 |
| RO3 | 3.70 | 4.70 | 5.80 | 0.75 | 2.3 | 1.56 | 2.90 |
| RS4 | 1.08 | 1.85 | 2.50 | 0.95 | 3.6 | 4.02 | 5.24 |
| RS5 | 2.80 | 3.60 | 4.30 | 1.25 | 3.4 | 3.62 | 3.25 |
| RS6 | 4.20 | 5.40 | 9.30 | 1.42 | 3.3 | 2.57 | 2.74 |

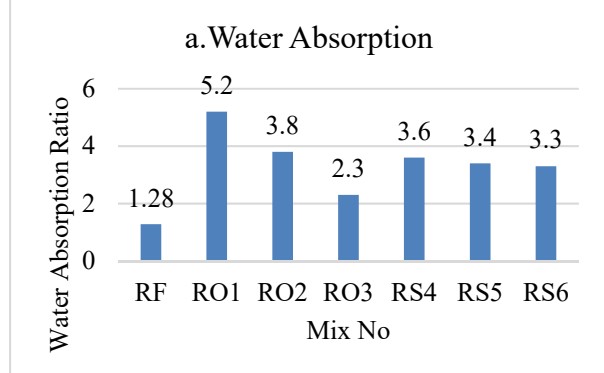

(**a**)

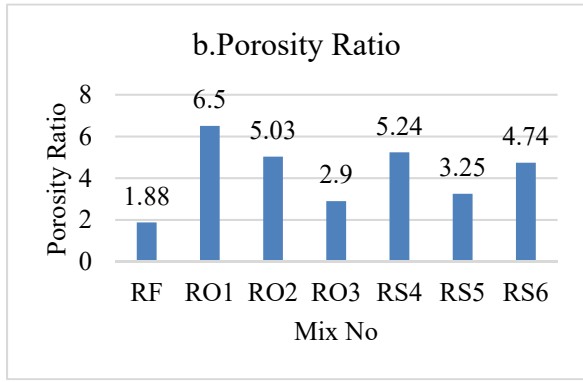

(**b**)

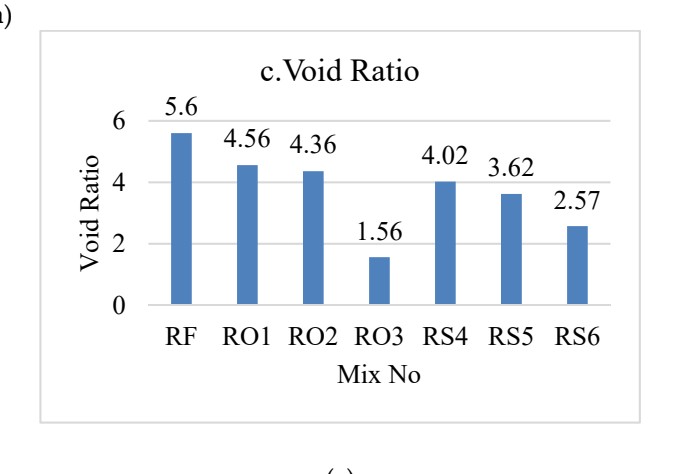

(**c**)

**Figure 6.** WRT concrete testing: (**a**) water absorption; (**b**) porosity; and (**c**) void ratio.

### 3.4. Compressive Strength

The most well-known and extensively used method of assessing compressive strength for quality control is the concrete cube test. Figure 7 shows the concrete compressive strength for all rubberized concrete samples after 7, 14, and 28 days. According to [13,16], the standard compressive strength of concrete can increase by 65% in 7 days, 90% in 14 days, and around 99% in 28 days.. The measurement values are presented in Table 7. The results demonstrate that the concrete compressive strength for OPC samples RO1, RO2,

and RO3 after 28 days is smaller than that of the reference sample RF by 84%, 70%, and 59%, respectively. In addition, the values for SRC samples RS4, RS5, and RS6 are smaller than those of the RF by 82%, 70%, and 34%, respectively. By comparison, for 50% WRT, the concrete compressive strength values after 28 days are higher for the RS6 mix with SRC, than for the RO3 mix with OPC. Therefore, the compressive strength of rubberized concrete can be increased to a greater extent by SRC than by OPC, although the percentage increase for RS6 is 25% higher than that of RO3. Therefore, according to the results, the most suitable percentage of WRT is 50% of RCA, which can more effectively enhance the rubberized concrete properties than other percentages (i.e., 100% and 75%).

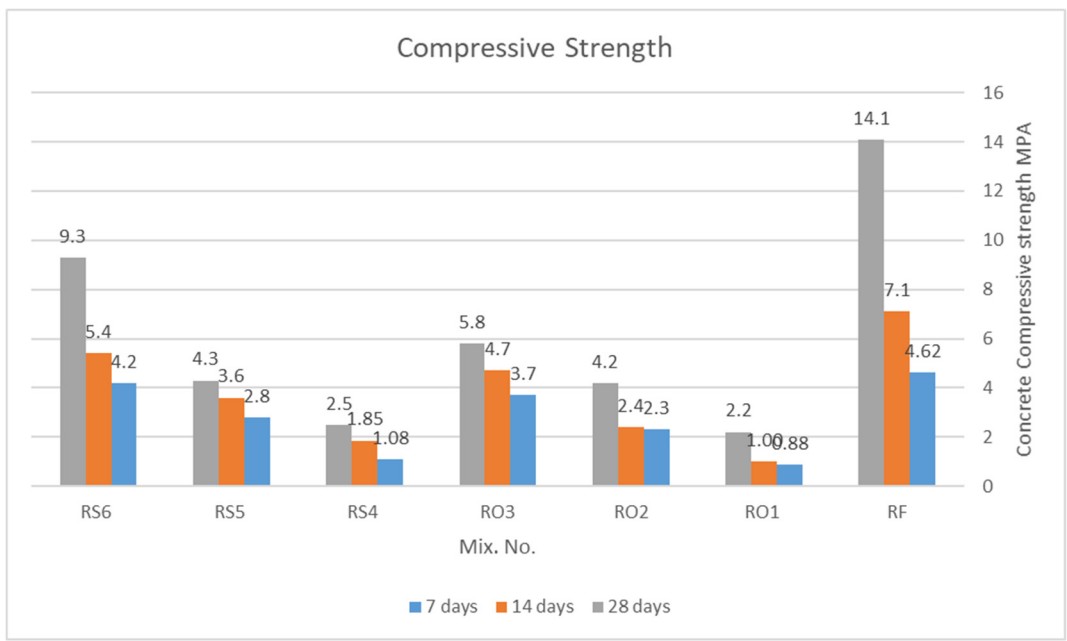

**Figure 7.** Compressive strength of different samples.

*3.5. Tensile Strength*

　　Table 7 and Figure 8 show the measurements of the splitting tensile strength test to investigate the effect of changing the type of cement on the characteristics of WRT. The results demonstrate the following.

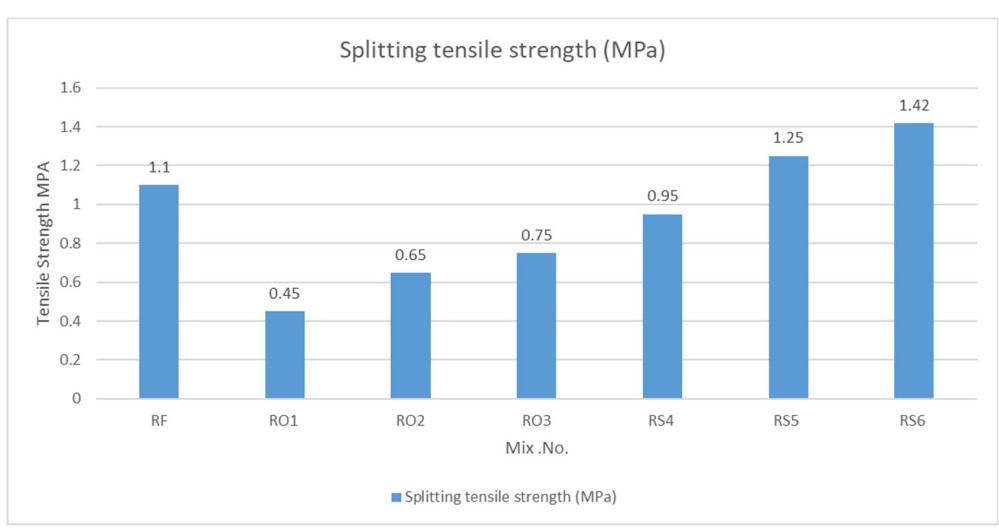

**Figure 8.** Splitting tensile strength.

The values for splitting tensile strength are lower for OPC samples RO1, RO2, and RO3 after 14 days, because the spike in the maximum strength gain is seen during the first 14 days; hence, we tested the concrete at intervals of 14 days only than those of the reference sample RF by 59%, 41%, and 32%, respectively.

For the SRC group, the values are higher for SRC samples RS5 and RS6 than for the RF by 14% and 29%, respectively. Therefore, the sulfate-resistant type of cement can increase the tensile strength of rubberized concrete to a greater extent than OPC. By comparison, for 50% WRT, the tensile strength is higher for the RS6 SRC mix than for the RO3 OPC mix by 89%. Hence, SRC can contribute to improving the characteristic of the tensile strength more than WRT, and this is related to the higher bond stresses between rubbers and cement for SRC.

### 3.6. Ductility Performance and Failure Mechanism

The capacity of a material to withstand considerable significant persistent deformation under a tensile load until it fractures is referred to as ductility. Therefore, ductility behavior is one of the important factors in determining the resistance in construction elements. For WRT, Zengh stated that rubberized concrete has lower brittleness index values than conventional concrete [1], and this is consistent with our results. The experiments also show the increase in ductility behavior, which is due to the extreme bonding between the wires in waste tires and cement. It is very difficult to evaluate ductility performance from only the results of crack failure behavior, and the ductility index evaluation method is based on a method that uses structural deflection and derivation of the energy area ratio. Ductility evaluation results are also based on the brittleness index values of rubberized concrete, which will also need to be further studied in the future.

In comparison to the RF concrete sample, the failure condition in rubberized concrete samples for OPC and SRC groups is characterized by higher deformation. This is due to the rubber chips, in addition to the failure state of the rubberized concrete, which takes a long time to progress. Additionally, the pozzolanic reaction between FA and calcium hydroxide can improve aggregate and matrix bonding properties with a higher plasticizer.

This study investigated the cracking behavior of both OPC and SRC. It can be stated that when the specimens reached the ultimate load, there were fewer large cracks in the rubberized concrete, although the increase in WRT led to brittleness of the concrete, which resulted in smaller cracks. By comparison, the ductility behavior of rubberized concrete was due to the extreme bonding between the wires in waste tires and cement. The OPC specimens were damaged earlier than the SRC specimens at the ultimate load, and specimen failure occurred as a result of spalling between the WRT and cement, which is indicated in Figure 7. Moreover, under a loading effect, the total volume of cracks changed according to the amount of WRT. The tests demonstrate that RS6 in the SRC group can improve the ductility and cracking resistance by more than RO3 in the OPC group. Furthermore, all samples with percentages of 50% WRT and 50% RCA have more ductility than others (75% and 100% WRT). In addition, Figure 9a shows the post-cracking behavior from the compression test.

Figure 9b demonstrates the shapes of failure for tensile splitting tests and shows plastic shrinkage cracking behavior stages for the RS6 specimen for the tensile test with SRC. This figure shows that the specimen cracked within the first 10 min of exposure, and shows the width of the specimen compared with a single crack in the sample with OPC. The width of the cracks for all the mixtures was measured at 3, 6, and 10 min. The results are cracks having an average width of about 1.2 and 0.6 mm at 3 min, 0.90 mm at 6 min, and, finally, 1.20 mm at 10 min; the total horizontal deformation is 15.50, 16.50, and 17.50 cm, respectively. In addition, Figure 9c shows the cracking failure and eccentricity direction with the effect of WRT in the split tensile test for sample RS6.

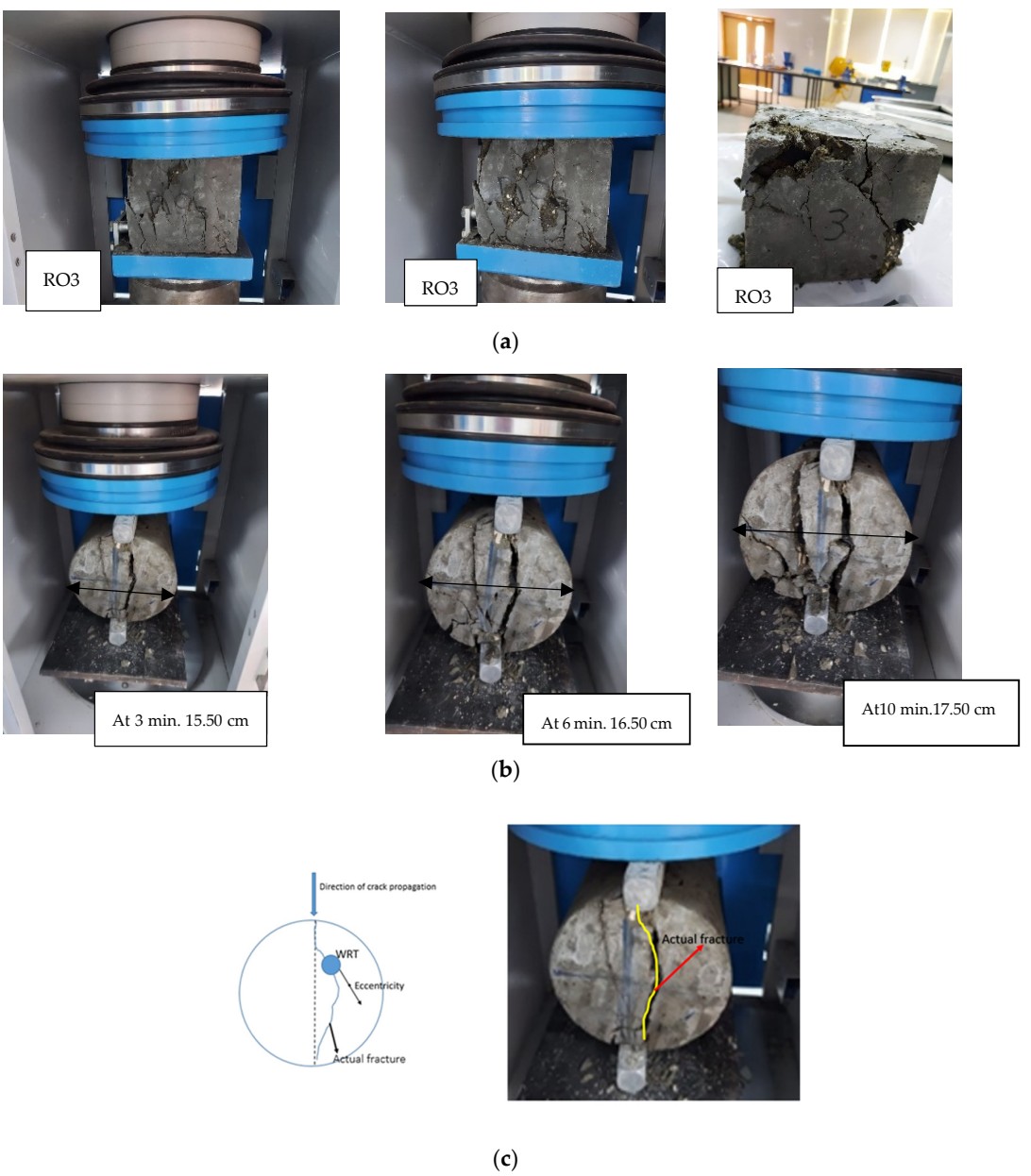

**Figure 9.** Cracking failure behavior for: (**a**) cubic compressive test (RO3); (**b**) split tensile test (RS6); (**c**) cracking failure and eccentricity effect of WRT split tensile test (RS6).

By comparison, rubberized concrete has lower brittleness index values than conventional concrete [11]. Our experiments showed the same result, whereby more ductile behavior was observed for rubberized concrete compared to the concrete specimen reference (RF) under compression and tensile testing. The failure state in tire rubber concrete compared to plain concrete is characterized by more deformation, and the failure state in rubberized concrete does not occur as quickly. Moreover, the pozzolanic reaction between FA and calcium hydroxide (C–H) can also form a stable hydrated calcium silicate (C–S–H), which can improve aggregate and matrix bonding properties.

From the experimental results, the fracture process in the concrete with different rubber contents under loading could be quantitatively compared. It was discovered that the inclusion of increased WRT resulted in increased brittleness of the concrete, leading to smaller cracks and less major cracks in the specimens when eventually loaded. However, the specimen was damaged due to the spalling between WRT and cement. Moreover, the changes in the total volume of the cracks under loading further suggest that, due to the

amount of rubber (WRT) replacing the recycled aggregate (RCA), in RS6 samples using SRC, the bond stresses and the resistance of cracking are further increased compared to RO3 having the same amount of WRT but a different type of cement (i.e., OPC). Consequently, more bonded behavior is observed for 50% WRT and 50% RCA than other samples.

### 3.7. Scanning Electron Microscopy (SEM)

SEM analysis was used for solid material characterization, where the signals generated during the analysis are transformed into two-dimensional pictures, revealing information about the investigated materials. The SEM investigation was conducted to study the interaction between the components of both OPC and SRC. From SEM, it is clear that silica fume (SF) and FA fill the spaces between cement and aggregate particles. Moreover, SF is considered to be a polymer material, which reacts with calcium hydroxide during the cement hydration process. These reactions lead to the formation of extra calcium silicate, hence resulting in increased density with reduced porosity. FA plays an important role in refining the internal structure because it sufficiently participates in the pozzolanic reaction and generates more C–S–H. Furthermore, concrete pores and cracks can be filled with C–S–H, which reduces the proportion of macropores and cracks, as shown by the results in [25]. This is also further evidence that the crystals grow in entrapped air voids instead of small pores, which happens at an early stage from 28 days [26]. In this process, the reduced overall crystallization stress can positively impact the sulfate resistance of rubberized specimens, and lead to more small cracks and microcracks appearing in SRC than in OPC specimens [14]. Therefore, there are increases in porosity and reductions in the concrete compressive strength of WRT, and cracking and crystal voids can be seen in the micrograph in Figure 10. Overall, SEM images show the presence of pores and voids around the rubber and the polymer composite, whereas uniform dispersion of FA and SF particles is observed in the polymer matrix. However, the SEM microstructural analyses suggest a higher proportion of C–S–H intermixed with the sulfate reaction phases of SRC rubberized mortar than those of OPC. Thus, with the growth in crystals in a decreased percentage of air voids, rather than a decrease in internal cracking, it is clearly shown that the average crack width is increased more in OPC mortar than in SRC.

More small cracks and microcracks appeared in SRC than OPC specimens. This is partly due to the increased porosity and reduced compressive strength of rubberized cement composites, which gradually expanded and became transfixed.

### 3.8. Statistical Analysis:

#### 3.8.1. *t*-Test Independent Method

*t*-testing was used as an inferential statistical tool to compare the mean and standard deviation of values of compressive and tensile strength for the two materials (OPC and SRC), to determine whether there is a sizeable distinction between the properties of these materials. In this case, a sizeable distinction means that the observed results are unlikely to be the result of chance or sampling mistakes. The *t*-test is one of many statistical tests that are used to test hypotheses. The *t*-test mathematically establishes the problem statement by choosing a sample from each of the two sets and assuming that the two means are equal. Using the appropriate formulas, certain values are calculated and compared to the standard values, and the putative null hypothesis is accordingly accepted or rejected. If the null hypothesis is rejected, it means that the data readings are strong and unlikely to be random. In general, a *t*-test requires three crucial data variables: the difference between mean values for each data group, the standard deviation for each group, and the number of data values for each group. A *t*-test, in essence, allows us to compare the average values of two datasets. The *t*-test can be used in our research to compare the mean and standard deviation of values of compressive and tensile strength for the two groups (OPC and SRC) to determine whether there is a significant difference between them. Moreover, in an experiment, a *t*-test can be used to determine whether differences between OPC and SRC groups are due to the manipulated variable or are simply due to chance. Therefore, the independent *t*-test

was used to determine whether the properties of concrete differed based on cement type; that is, the *t*-test was used to determine if the difference between the groups is meaningful or random.

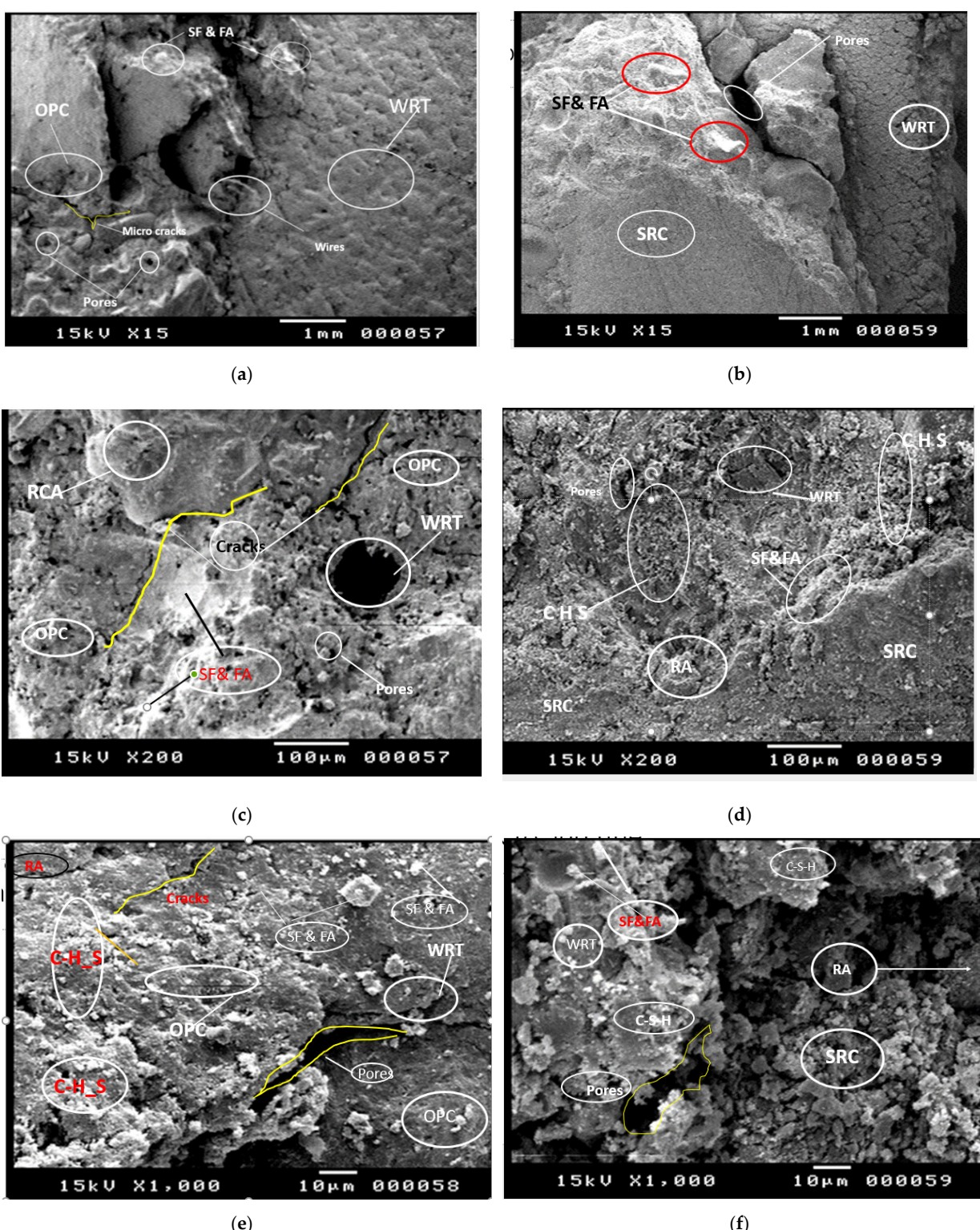

**Figure 10.** SEM micrographs of rubberized concrete: (**a**) (RO1) OPC with (100%WRT + 0%RCA); (**b**) (RS4) SRC with (100%WRT + 0%RCA); (**c**) (RO2) OPC with (75%WRT + 25%RCA); (**d**) (RS5) SRC with (75%WRT + 25%RCA); (**e**) (RO3) OPC with (50%WRT + 50%RA); (**f**) (RS6) SRC with (50%WRT + 50%RA).

When the number of samples in each group is the same or the variance of the two datasets is similar, the equal variance *t*-test is applied. For an equal variance *t*-test, the following formula is used to obtain the t-value and degrees of freedom:

$$t_{value} = \frac{Mean1 - Mean2}{\frac{(n1-1)\times var_1^2 + (n2-1)\times var_2^2}{n1+n2-2} \times \sqrt{\frac{1}{n1} + \frac{1}{n2}}} \tag{1}$$

where:

*Mean* 1 and *Mean* 2 are the average values for groups 1 and 2, respectively;
*var*. 1 and *var*. 2 are the variances for groups 1 and 2, respectively;
*n*1 and *n*2 are the numbers of samples for groups 1 and 2, respectively.

The average results of compressive strength for the rubberized concrete at 7 and 28 days were analyzed using the *t*-test. This can also be used for the tensile splitting test, and the independent *t*-test can be used to determine the difference between the measurements of OPC and SRC groups.

The results were analyzed using a statistical *t*-test, which includes two primary hypotheses: the null hypothesis and the alternative hypothesis. The null hypothesis indicates that the strength values from OPC and SRC are not significantly different. Conversely, the alternative hypothesis displays a substantial difference in data, and the goal of the findings of the statistical analysis is to determine the degree of difference in the results of matched experiments. When *p*-values are greater than 0.05, the null hypothesis is accepted; conversely, when *p*-values are less than 0.05, the alternative hypothesis is accepted.

Tables 8–15 show the independent *t*-test results, which indicate if there are significant differences and whether or not to accept or reject the null hypothesis. Table 16 shows the findings of the statistical analysis of the compressive strength of both types of concrete after 7, 14, and 28 days, and the splitting tensile strength. The following can be stated:

- For concrete compressive strength after 7, 14, and 28 days, the findings of $F_{cu}$ testing for OPC and SRC cement types show no significant differences.
- For concrete splitting tensile strength after 14 days, the null hypothesis was rejected, and the alternative hypothesis is preferred; therefore, there is a significant difference between the results of tensile strength tests for OPC and SRC.

**Table 8.** Group statistics for concrete compressive strength test after 7 days.

| Type of Test | Groups | N | Mean | SD |
| --- | --- | --- | --- | --- |
| Concrete compressive strength after 7 days | OPC group | 3.0 | 2.2933 | 1.41001 |
| | SRC group | 3.0 | 2.6933 | 1.56273 |

**Table 9.** Independent *t*-test for concrete compressive strength after 7 days.

| | | Levene's Test for Equality of Variances | | *t*-Test for Equality of Means | | | | | | |
| --- | --- | --- | --- | --- | --- | --- | --- | --- | --- | --- |
| | | | | | | | | | 95% Confidence Interval of the Difference | |
| | | F | Sig | t | df | Sig. (2-Tailed) | Mean Difference | Std. Error Difference | Lower | Upper |
| Concrete compressive strength after 7 Days | Equal variances assumed | 0.039 | 0.853 | −0.329 | 4.0 | 0.759 | −0.40000 | 1.21522 | −3.77399 | 2.97399 |
| | Equal variances not assumed | - | - | −0.329 | 3.958 | 0.759 | −0.40000 | 1.21522 | −3.78801 | 2.98801 |

**Table 10.** Group statistics for concrete compressive strength test after 14 days.

| Type of Test | Groups | N | Mean | Std. Deviation |
|---|---|---|---|---|
| Concrete compressive strength after 14 days | OPC group | 3.0 | 2.7000 | 1.86815 |
| | SRC group | 3.0 | 3.6167 | 1.77506 |

**Table 11.** Independent *t*-test for concrete compressive strength after 14 days.

| | | Levene's Test for Equality of Variances | | *t*-Test for Equality of Means | | | | | | |
|---|---|---|---|---|---|---|---|---|---|---|
| | | | | | | | | | 95% Confidence Interval of the Difference | |
| | | F | Sig | t | df | Sig. (2-Tailed) | Mean Difference | Std. Error Difference | Lower | Upper |
| Concrete compressive strength after 14 days | Equal variances assumed | 0.034 | 0.863 | −0.616 | 4.0 | 0.571 | −0.91667 | 1.48782 | −5.04752 | 3.21419 |
| | Equal variances not assumed | - | - | −0.616 | 3.990 | 0.571 | −0.91667 | 1.48782 | −5.05177 | 3.21844 |

**Table 12.** Group statistics for concrete compressive strength test after 28 days.

| Type of Test | Groups | N | Mean | Std. Deviation |
|---|---|---|---|---|
| Concrete compressive strength after 28 days | OPC group | 3.0 | 4.0667 | 1.80370 |
| | SRC group | 3.0 | 5.3667 | 3.52326 |

**Table 13.** Independent *t*-test for concrete compressive strength after 28 days.

| | | Levene's Test for Equality of Variances | | *t*-Test for Equality of Means | | | | | | | |
|---|---|---|---|---|---|---|---|---|---|---|---|
| | | | | | | | | | .5 | 95% Confidence Interval of the Difference | |
| | | F | Sig. | t | df | Sig. (2-Tailed) | Mean Difference | Std. Error Difference | | Lower | Upper |
| Concrete compressive strength after 28 days | Equal variances assumed | 1.880 | 0.242 | −0.569 | 4.0 | 0.600 | −1.30000 | 2.28522 | | −7.64478 | 5.04478 |
| | Equal variances not assumed | - | - | −0.569 | 2.981 | 0.609 | −1.30000 | 2.28522 | | −8.59894 | 5.99894 |

**Table 14.** Group statistics for splitting tensile strength test after 14 days.

| Type of Test | Groups | N | Mean | Std. Deviation |
|---|---|---|---|---|
| Splitting tensile strength | OPC group | 3.0 | 0.6167 | 0.15275 |
| | SRC group | 3.0 | 1.2067 | 0.23798 |

**Table 15.** Independent *t*-test for splitting tensile strength after 14 days.

| | | Levene's Test for Equality of Variances | | *t*-Test for Equality of Means | | | | | 95% Confidence Interval of the Difference | |
| --- | --- | --- | --- | --- | --- | --- | --- | --- | --- | --- |
| | | F | Sig | t | df | Sig. (2-Tailed) | Mean Difference | Std. Error Difference | Lower | Upper |
| Splitting tensile strength after 14 days | Equal variances assumed | 0.616 | 0.476 | −3.614 | 4.0 | 0.022 | −0.59000 | 0.16327 | −1.04330 | −0.13670 |
| | Equal variances not assumed | - | - | −3.614 | 3.409 | 0.029 | −0.59000 | 0.16327 | −1.07605 | −0.10395 |

**Table 16.** Results and conclusion of the independent *t*-test.

| Type of Test | Significance Level α Versus *p*-Value | Results |
| --- | --- | --- |
| Concrete compressive strength after 7 days | *p*-value = 0.759<br>α = 0.05<br>*p*-value > α;<br>The null hypothesis is accepted, and an alternative hypothesis is unable to replace it. | The findings of concrete compressive strength testing for OPC and SRC cement types show no significant differences, and the SRC yielded an average strength of 1.17 of the compressive strength of the OPC group. |
| Concrete compressive strength after 14 days | *p*-value = 0.571<br>α = 0.05<br>*p*-value > α;<br>The null hypothesis is accepted, and an alternative hypothesis is unable to replace it. | The findings of concrete compressive strength testing for OPC and SRC cement types show no significant differences, and the SRC yielded an average strength of 1.33 of the compressive strength of the OPC group. |
| Concrete compressive strength after 28 days | *p*-value = 0.600<br>α = 0.05<br>*p*-value > α;<br>The null hypothesis is accepted, and an alternative hypothesis is unable to replace it. | The findings of concrete compressive strength testing for OPC and SRC cement types show no significant differences, and the SRC yielded an average strength of 1.31 of the compressive strength of the OPC group. |
| Concrete splitting tensile strength after 14 days | *p*-value = 0.022<br>α = 0.05<br>*p*-value < α;<br>The null hypothesis is rejected, and the alternative hypothesis is preferred. | There is a significant difference between the results of tensile strength for OPC and SRC, and the average strength obtained from the SRC is equivalent to 1.95 of the average concrete splitting tensile strength of the OPC group. |

### 3.8.2. Relationship between Porosity and Bulk Density

Regression analysis can be used to create a correlation curve between the variable values of porosity and bulk density for both the OPC and SRC groups. Figure 11 indicates the correlation curve between them and the value of $R^2$ for both groups (OPC and SRC), which indicates an increase in the convergence between the values of the correlation curves.

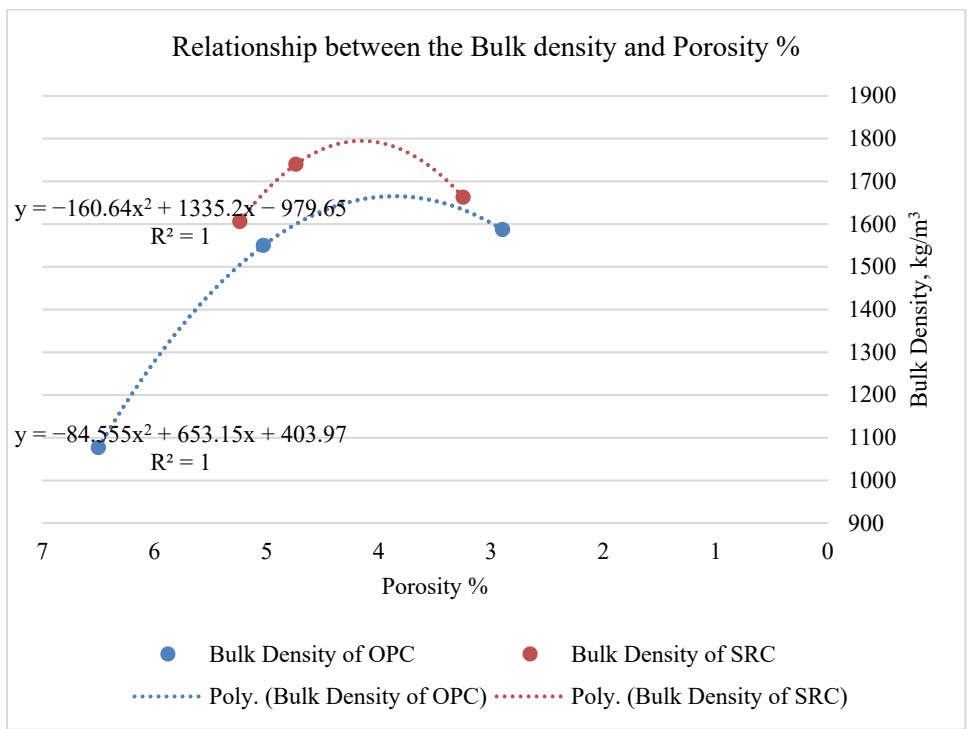

**Figure 11.** Differences between the values of OPC and SRC for bulk density.

Based on the relationships shown in Figure 11, the differences between the values of OPC and SRC are significantly increased, by 49% for bulk density, for 100% WRT. The porosity ratio varies by nearly 19%, and the variations between the types (OPC and SRC) for other samples are between 7% and 10% for both 75% and 50% WRT, with variations in the porosity ratio between 35% and 63%. By comparison, with increasing WRT, the porosity ratio and the void ratio increase, implying that the concrete permeability increases with an increase in porosity and a decrease in density. Additionally, porosity is the primary factor that governs the strength of brittle materials. As the capillary porosity decreases, compressive strength increases. However, this is mainly dependent on the cement type, with OPC resulting in a higher porosity ratio than SRC. In general, the porosity is increased with increasing WRT content. As the porosity ratio is higher with OPC than with SRC, this leads to a greater likelihood of failure due to the brittleness of the concrete, as in the case of RO1. The findings demonstrate that:

1.  The increase in slump for different percentages of WRT in the samples leads to increased workability for OPC and SRC because the water absorption of WRT increases.
2.  The density values of WRT for the two types of cement match with the range of densities in lightweight concrete, where LWC is defined according to ACI 213.
3.  There are noticeable decreases in the void ratios for rubberized concrete prepared from both OPC and SRC mixtures.
4.  The SRC type can increase the compressive strength of rubberized concrete, which occurs more dramatically than for OPC. However, SRC can also contribute more greatly to improving the characteristic of tensile strength compared with OPC, and this is related to the higher bond stresses between WRT and SRC.
5.  When the amount of WRT is increased in the mixture, the brittleness of the concrete increases, resulting in smaller cracks when eventually loaded. The specimen with OPC has more brittleness and is eventually destroyed earlier than SRC at the ultimate load.
6.  SEM images show the presence of pores and voids around the rubber and the polymer composite, whereas uniform dispersion of FA and SF particles is observed in the polymer matrix. However, SEM microstructural analyses suggest a higher proportion of C–S–H intermixed with sulfate reaction phases of SRC rubberized mortar than with

OPC. Thus, there is crystal growth in a decreased percentage of air voids, rather than a decrease in internal cracking.

7. It is clearly shown that there is a greater increase in the average crack width for OPC mortar than for SRC. More small cracks and microcracks appeared in SRC than OPC specimens. This is partly due to the increased porosity and reduced compressive strength of rubberized cement composites, which gradually expanded and became transfixed.

## 4. Conclusions

This study investigated the effects of cement types used within a rubberized concrete mixture on the resulting properties. The most important aspects that were affected are workability, compressive strength, split tensile, ductility, and failure behavior. The waste tires rubber aggregate was mixed with either SRC or OPC. All mixtures contained 10% SF and 0.2% fly ash, and the percentage of WRT content was varied between 0%, 50%, 75%, and 100%. Overall, the results indicate the following:

- The workability was higher for fresh rubberized mixtures of SRC than those of OPC.
- Increasing the amount of WRT as a replacement for RCA caused a decrease in the density by approx. 50.5% for OPC and 40% for SRC, although the highest increase in porosity was observed, of approx. 56%, for both types of cement.
- The reductions in WRT as a percentage of RCA led to decreases in water absorption (WAR) and void ratios for both OPC and SRC; in contrast, water absorption and air voids were higher in OPC than in SRC, by 30% and 13%, respectively.
- WRT rubber reduced the compressive and tensile strength by 35% and 60%, respectively, for both SRC and OPC, and the reduction in the amount of WRT used to replace RCA concrete also led to increased compressive and tensile strength. Therefore, the decreasing percentage depends on the amount of recycled rubber in the concrete mixture.
- Recycled rubber positively contributed to the reduction in cracks in the resulting concrete.
- Replacement WRT in concrete 50% from weight of coarse aggregate had the effect of increasing the ductility and subsequently reducing brittleness as a factor of.
- SEM images showed the presence of pores/voids around the rubber and the polymer composite, whereas uniform dispersion of silica fume particles was observed in the polymer matrix.
- The SEM microstructural analysis showed a higher proportion of C–S–H intermixed with the sulfate reaction phases of SRC rubberized mortar than with OPC; thus, crystal growth in a decreased percentage of air voids resulted in a greater increase in the average crack width for OPC mortar than that of SRC.
- The independent *t*-test was used to compare the average values of compressive and tensile strengths for OPC and SRC, and the findings demonstrate that the differences in $F_{cu}$ between the OPC and SRC groups were not significant; this means the null hypothesis was accepted according to the statistical analysis. In contrast, for tensile strength (Ft), the null hypothesis was rejected, and the alternative hypothesis was preferred; therefore, there was a significant difference between the results of tensile strength for OPC and SRC.
- For the correlation curve between porosity and bulk densities for both OPC and SRC, the values of $R^2$ for both the OPC and SRC groups were 1.0; therefore, the values were very close in the correlation curves.
- As a recommendation for future work, study should be undertaken of the ductility index evaluation method and the derivation of the energy area ratio, to ensure that ductility evaluation results are based on the brittleness index values of rubberized concrete.

**Author Contributions:** L.K.I.: conceptualization, methodology, investigation, writing—review and editing of the original draft and submission; Y.A.S.G.: acquisition of data, analysis and interpreta-tion, and critical revision. All authors have read and agreed to the published version of the manuscript.

**Funding:** This research received no external funding.

**Institutional Review Board Statement:** Not applicable.

**Informed Consent Statement:** Not applicable.

**Data Availability Statement:** All data and material are included in the article.

**Acknowledgments:** We appreciate Sphinx University for their assistance in supplying us with all possible options in the concrete structure laboratory.

**Conflicts of Interest:** The authors declare no conflict of interest.

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
