# Peer review of "Properties of Rubberized Concrete Prepared from Different Cement Types"

_recycling, doi:10.3390/recycling7030039_

Round 1

Reviewer 1 Report

The research of the paper is relatively novel, but the format of the paper is very irregular with a large error rate, for example, the initial letter of “moreover” should be capitalized in the second line of the abstract section; Keywords should be separated by semicolons; the outer frame of the figures should be hidden, etc. Therefore it will affect the reader's correct understanding of the text. It is recommended to overhaul the paper before considering whether to accept it or not.

Author Response

Dear  Professor (Reviewer)

Greetings

 We would like to thank you for the letter dated 13/5/2022, and for the opportunity to resubmit a revised copy of this manuscript.

We would also like to take this opportunity to express our thanks to the reviewers for the positive feedback and helpful comments regarding the correction or modification of our manuscript titled (Rubberized concrete properties with different cement types).
We believe have resulted in an improved revised manuscript, which you will find uploaded alongside this document.

About your request for correcting an overhaul of the paper

The response: Many thanks for your comments and the overhaul article has been corrected.  And  our manuscript has be sent for editing language check the (MDPI confirms that it has received payment of English editing invoiceenglish-44639 (invoice dated18 May 2022)

We very much hope the revised manuscript is accepted for publication in your respected Recycling journal

Thanks

Lamiaa K Idriss

The corresponding author

Reviewer 2 Report

This study investigated the effects of the waste tire particles on the properties of different types of cement. The waste rubber tire particles were used to replace 50%, 75%, and 100% of recycled concrete, and two types of cement (OPC and SRC) were applied in this study. The results showed that the addition of the WRT increased the workability, water absorption, and porosity ratio and reduced the compressive as well as tensile strength. However, the effects of the rubber on the properties of cementitious materials have been well documented in the literature, especially for the reported properties. The reported results in this manuscript are generally consistent with that in other literature. The contributions of this study to this research field are unclear. In addition, the discussion part of this manuscript lacks deep analysis and is more like a report rather than a scientific article. The significance and the novelty of this study should be demonstrated and the discussion part should be improved.

Author Response

Dear  Professor (Reviewer)

Greetings

 We would like to thank you for the letter dated 13/5/2022, and for the opportunity to resubmit a revised copy of this manuscript.

We would also like to take this opportunity to express our thanks to the reviewers for the positive feedback and helpful comments regarding the correction or modification of our manuscript titled (Rubberized concrete properties with different cement types).
We believe have resulted in an improved revised manuscript, which you will find uploaded alongside this document.

1-About Your request for The contributions of this study to this research, the field is unclear.

The response (1): Many thanks for your valuable comments and the contribution of our study has been added, check page 3 (line 29 to line 31)

2- the discussion part of this manuscript lacks deep analysis. The significance and the novelty of this study should be demonstrated.

The response(2):

Many thanks for your valuable comments and the significance and the novelty of our study have been added, check page 22 (line 11 to line 22) and check page 23 (line 1 to line 7).  

And  our manuscript has been sent for editing language check the (MDPI confirms that it has received payment of English editing invoiceenglish-44639 (invoice dated18 May 2022

We very much hope the revised manuscript is accepted for publication in your respected Recycling journal

Thanks

Lamiaa K Idriss

The corresponding author

Reviewer 3 Report

This paper focuses on analyzing the effect of using waste tires rubber instead of recycled aggregate as coarse aggregate on two types of cement (sulfate resistance cement and ordinary Portland cement). The topic addressed is interesting and deserves a constructive discussion for industrial waste reuse and recycling. However, several points as indicated below need to be addressed by authors to improve the quality of the article.

1. p.1, Abstract Line19

Do “SRC” and “SCR” mean the same thing? If that’s so, please use the same term throughout the paper.

2. p.3, Line 4 from the bottom

I don’t understand what “20mm” means. Does the 20mm have the same meaning as maximum size of aggregate? Could you please clarify what you mean by this?

3. p.3, Line 1 from the bottom

Please indicate the physical properties (density and water absorption) of waste tires. These values are important for confirming mix proportion of concrete.

4. p.4, 2.2 Cement

Please explain why you used sulfate resistance cement. And please also explain the reason for adding silica fume and fly ash.

5. p.5, Line 7 from the bottom

“binder content of 507 kg/m3”

In Table 5, cement weight is 396 kg/m3. Which value is correct?

6. p.5, Line 6 from the bottom

“2% (by weight) replacement”

In Table 5, FA% of the cement content is 0.20%. Which value is correct?

7. p.8, Line 8 from the bottom

Please explain the reason the slump increases with adding WRT.

8. p.9, Line 3 from the bottom

“the disadvantage of lowering air content may lead to reduce some of the desired effects of tire rubber in concrete”

What does the desired effects mean? I don’t understand it.

9. p.12, 3.6 Ductility performance and Failure Mechanism

I think it is very difficult to evaluate ductility performance only from the result of crack failure behavior. Quantitative evaluation using indicators such as stress-strain curve and fracture energy is necessary.

10. p.13, Fig.9

Please add the name of the specimen (mix No.) in this photo. And the photo at the bottom is the result of tensile test, so please correct the sub-caption.

Furthermore, please add the meaning of the values in the bottom of photo.

11. p.14, 3.7 SEM

The discussion on the SEM images is very confusing. For example,

“it is clear that silica fume and fly ash the spaces between cement and aggregate particles,”

What part of the SEM image can be determined to be cement and aggregates particles? Then, please point out the area where silica fume and fly ash are filling the void in SEM image.

12. p.18, 3.8.2 The relationship between porosity and the bulk density

I don’t understand what can be explained by this relationship. Please add in detail.

Author Response

Dear  Professor (Reviewer)

Greetings

 We would like to thank you for the letter dated 13/5/2022, and for the opportunity to resubmit a revised copy of this manuscript.

We would also like to take this opportunity to express our thanks to the reviewers for the positive feedback and helpful comments regarding the correction or modification of our manuscript titled (Rubberized concrete properties with different cement types).
We believe have resulted in an improved revised manuscript, which you will find uploaded alongside this document.

1- The reviewer question:

  1. p.1, Abstract Line19 Do “SRC” and “SCR” mean the same thing? If that’s so, please use the same term throughout the paper.

The response (1): Many thanks for your valuable comments. Yes,  It is the same meaning, we have amended the wrong term in our manuscript.

2- The reviewer question:

2- p.3, Line 4 from the bottom. I don’t understand what “20mm” means. Does the 20mm have the same meaning as the maximum size of aggregate? Could you please clarify what you mean by this?

The response (2): Many thanks for your valuable comments. In recycling coarse aggregates (RCA) that had been used in the experiments maximum size of aggregate is 20 mm.

3- The reviewer question:

  1. p.3, Line 1 from the bottom

Please indicate the physical properties (density and water absorption) of waste tires. These values are important for confirming mix proportion of concrete.

The response (3): Many thanks for your appreciated comments

Your request has been done please  kindly check page 4 (Table 1:Coarse and fine Aggregates properties)

4- The reviewer question:

4-p.4, 2.2 Cement

Please explain why you used sulfate resistance cement. And please also explain the reason for adding silica fume and fly ash.

The response (4): Many thanks for your respected comments. Kindly check page (5) 

 5- The reviewer question:

  1. p.5, Line 7 from the bottom “binder content of 507 kg/m3”

In Table 5, cement weight is 396 kg/m3. Which value is correct?

 The response (5): cement weight is 396 kg/m3 was corrected

6- The reviewer Question  

  1. p.5, Line 6 from the bottom

“2% (by weight) replacement”

In Table 5, FA% of the cement content is 0.20%. Which value is correct?

The Response (6): FA% of the cement content is 0.20%  has been corrected

7. p.8, Line 8 from the bottom‎

7- The reviewer Question  Please explain the reason the slump increases with adding WRT.‎

 The response (7): The slump increases with adding SF and fly ash at WRT at [OPC]and [SRC] because ‎ water ‎absorption of WRT increases, The initial porosity of WRT increases compared to the ‎reference RF, and the density decreased. ‎

8- The reviewer question 8

8. p.9, Line 3 from the bottom‎

‎“The disadvantage of lowering air content may lead to reduce some of the desired effects of ‎tire rubber in concrete”‎

What does the desired effects mean? I don’t understand it.‎

The response(8): That means air-entrained concrete with low content of air voids. Optimal air entrainment ‎will protect concrete from the harmful effects of the interaction of sulfate attack and ‎freeze-thaw damage.‎

9- The reviewer question 9

P.12, 3.6 Ductility performance and Failure Mechanism

I think it is very difficult to evaluate ductility performance only from the result of crack failure behavior. Quantitative evaluation using indicators such as stress-strain curve and fracture energy is necessary.

The Response 9 : Thanks for your comment your request has been added please check page 13,14)

10- The reviewer Question 10

  1. p.13, Fig.9 Please add the name of the specimen (mix No.) in this photo. And the photo at the bottom is the result of tensile test, so please correct the sub-caption.

Furthermore, please add the meaning of the values in the bottom of photo.

Response 10: Your request has been done check page 14, 15 (fig 9)

  1. The reviewer question 11

‎ 11. p.14, 3.7 SEM .The discussion on the SEM images is very confusing.

Response 11:  kindly check fig 10 and  page 16-18

12.The reviewer question 12

12. p.18, 3.8.2 The relationship between porosity and the bulk density

I don’t understand what can be explained by this relationship. Please add in detail.‎

Response 12:  kindly check page 22

And  our manuscript has been sent for editing language check the (MDPI confirms that it has received payment of English editing invoiceenglish-44639 (invoice dated18 May 2022

We very much hope the revised manuscript is accepted for publication in your respected Recycling journal

Thanks

Lamiaa K Idriss

The corresponding author

Dear  Professor (Reviewer)

Greetings

 We would like to thank you for the letter dated 13/5/2022, and for the opportunity to resubmit a revised copy of this manuscript.

We would also like to take this opportunity to express our thanks to the reviewers for the positive feedback and helpful comments regarding the correction or modification of our manuscript titled (Rubberized concrete properties with different cement types).
We believe have resulted in an improved revised manuscript, which you will find uploaded alongside this document.

1- The reviewer question:

  1. p.1, Abstract Line19 Do “SRC” and “SCR” mean the same thing? If that’s so, please use the same term throughout the paper.

The response (1): Many thanks for your valuable comments. Yes,  It is the same meaning, we have amended the wrong term in our manuscript.

2- The reviewer question:

2- p.3, Line 4 from the bottom. I don’t understand what “20mm” means. Does the 20mm have the same meaning as the maximum size of aggregate? Could you please clarify what you mean by this?

The response (2): Many thanks for your valuable comments. In recycling coarse aggregates (RCA) that had been used in the experiments maximum size of aggregate is 20 mm.

3- The reviewer question:

  1. p.3, Line 1 from the bottom

Please indicate the physical properties (density and water absorption) of waste tires. These values are important for confirming mix proportion of concrete.

The response (3): Many thanks for your appreciated comments

Your request has been done please  kindly check page 4 (Table 1:Coarse and fine Aggregates properties)

4- The reviewer question:

4-p.4, 2.2 Cement

Please explain why you used sulfate resistance cement. And please also explain the reason for adding silica fume and fly ash.

The response (4): Many thanks for your respected comments. Kindly check page (5) 

 5- The reviewer question:

  1. p.5, Line 7 from the bottom “binder content of 507 kg/m3”

In Table 5, cement weight is 396 kg/m3. Which value is correct?

 The response (5): cement weight is 396 kg/m3 was corrected

6- The reviewer Question  

  1. p.5, Line 6 from the bottom

“2% (by weight) replacement”

In Table 5, FA% of the cement content is 0.20%. Which value is correct?

The Response (6): FA% of the cement content is 0.20%  has been corrected

7. p.8, Line 8 from the bottom‎

7- The reviewer Question  Please explain the reason the slump increases with adding WRT.‎

 The response (7): The slump increases with adding SF and fly ash at WRT at [OPC]and [SRC] because ‎ water ‎absorption of WRT increases, The initial porosity of WRT increases compared to the ‎reference RF, and the density decreased. ‎

8- The reviewer question 8

8. p.9, Line 3 from the bottom‎

‎“The disadvantage of lowering air content may lead to reduce some of the desired effects of ‎tire rubber in concrete”‎

What does the desired effects mean? I don’t understand it.‎

The response(8): That means air-entrained concrete with low content of air voids. Optimal air entrainment ‎will protect concrete from the harmful effects of the interaction of sulfate attack and ‎freeze-thaw damage.‎

9- The reviewer question 9

P.12, 3.6 Ductility performance and Failure Mechanism

I think it is very difficult to evaluate ductility performance only from the result of crack failure behavior. Quantitative evaluation using indicators such as stress-strain curve and fracture energy is necessary.

The Response 9 : Thanks for your comment your request has been added please check page 13,14)

10- The reviewer Question 10

  1. p.13, Fig.9 Please add the name of the specimen (mix No.) in this photo. And the photo at the bottom is the result of tensile test, so please correct the sub-caption.

Furthermore, please add the meaning of the values in the bottom of photo.

Response 10: Your request has been done check page 14, 15 (fig 9)

  1. The reviewer question 11

‎ 11. p.14, 3.7 SEM .The discussion on the SEM images is very confusing.

Response 11:  kindly check fig 10 and  page 16-18

12.The reviewer question 12

12. p.18, 3.8.2 The relationship between porosity and the bulk density

I don’t understand what can be explained by this relationship. Please add in detail.‎

Response 12:  kindly check page 22

And  our manuscript has been sent for editing language check the (MDPI confirms that it has received payment of English editing invoiceenglish-44639 (invoice dated18 May 2022

We very much hope the revised manuscript is accepted for publication in your respected Recycling journal

Thanks

Lamiaa K Idriss

The corresponding author

Dear  Professor (Reviewer)

Greetings

 We would like to thank you for the letter dated 13/5/2022, and for the opportunity to resubmit a revised copy of this manuscript.

We would also like to take this opportunity to express our thanks to the reviewers for the positive feedback and helpful comments regarding the correction or modification of our manuscript titled (Rubberized concrete properties with different cement types).
We believe have resulted in an improved revised manuscript, which you will find uploaded alongside this document.

1- The reviewer question:

  1. p.1, Abstract Line19 Do “SRC” and “SCR” mean the same thing? If that’s so, please use the same term throughout the paper.

The response (1): Many thanks for your valuable comments. Yes,  It is the same meaning, we have amended the wrong term in our manuscript.

2- The reviewer question:

2- p.3, Line 4 from the bottom. I don’t understand what “20mm” means. Does the 20mm have the same meaning as the maximum size of aggregate? Could you please clarify what you mean by this?

The response (2): Many thanks for your valuable comments. In recycling coarse aggregates (RCA) that had been used in the experiments maximum size of aggregate is 20 mm.

3- The reviewer question:

  1. p.3, Line 1 from the bottom

Please indicate the physical properties (density and water absorption) of waste tires. These values are important for confirming mix proportion of concrete.

The response (3): Many thanks for your appreciated comments

Your request has been done please  kindly check page 4 (Table 1:Coarse and fine Aggregates properties)

4- The reviewer question:

4-p.4, 2.2 Cement

Please explain why you used sulfate resistance cement. And please also explain the reason for adding silica fume and fly ash.

The response (4): Many thanks for your respected comments. Kindly check page (5) 

 5- The reviewer question:

  1. p.5, Line 7 from the bottom “binder content of 507 kg/m3”

In Table 5, cement weight is 396 kg/m3. Which value is correct?

 The response (5): cement weight is 396 kg/m3 was corrected

6- The reviewer Question  

  1. p.5, Line 6 from the bottom

“2% (by weight) replacement”

In Table 5, FA% of the cement content is 0.20%. Which value is correct?

The Response (6): FA% of the cement content is 0.20%  has been corrected

7. p.8, Line 8 from the bottom‎

7- The reviewer Question  Please explain the reason the slump increases with adding WRT.‎

 The response (7): The slump increases with adding SF and fly ash at WRT at [OPC]and [SRC] because ‎ water ‎absorption of WRT increases, The initial porosity of WRT increases compared to the ‎reference RF, and the density decreased. ‎

8- The reviewer question 8

8. p.9, Line 3 from the bottom‎

‎“The disadvantage of lowering air content may lead to reduce some of the desired effects of ‎tire rubber in concrete”‎

What does the desired effects mean? I don’t understand it.‎

The response(8): That means air-entrained concrete with low content of air voids. Optimal air entrainment ‎will protect concrete from the harmful effects of the interaction of sulfate attack and ‎freeze-thaw damage.‎

9- The reviewer question 9

P.12, 3.6 Ductility performance and Failure Mechanism

I think it is very difficult to evaluate ductility performance only from the result of crack failure behavior. Quantitative evaluation using indicators such as stress-strain curve and fracture energy is necessary.

The Response 9 : Thanks for your comment your request has been added please check page 13,14)

10- The reviewer Question 10

  1. p.13, Fig.9 Please add the name of the specimen (mix No.) in this photo. And the photo at the bottom is the result of tensile test, so please correct the sub-caption.

Furthermore, please add the meaning of the values in the bottom of photo.

Response 10: Your request has been done check page 14, 15 (fig 9)

  1. The reviewer question 11

‎ 11. p.14, 3.7 SEM .The discussion on the SEM images is very confusing.

Response 11:  kindly check fig 10 and  page 16-18

12.The reviewer question 12

12. p.18, 3.8.2 The relationship between porosity and the bulk density

I don’t understand what can be explained by this relationship. Please add in detail.‎

Response 12:  kindly check page 22

And  our manuscript has been sent for editing language check the (MDPI confirms that it has received payment of English editing invoiceenglish-44639 (invoice dated18 May 2022

We very much hope the revised manuscript is accepted for publication in your respected Recycling journal

Thanks

Lamiaa K Idriss

The corresponding author

Dear  Professor (Reviewer)

Greetings

 We would like to thank you for the letter dated 13/5/2022, and for the opportunity to resubmit a revised copy of this manuscript.

We would also like to take this opportunity to express our thanks to the reviewers for the positive feedback and helpful comments regarding the correction or modification of our manuscript titled (Rubberized concrete properties with different cement types).
We believe have resulted in an improved revised manuscript, which you will find uploaded alongside this document.

1- The reviewer question:

  1. p.1, Abstract Line19 Do “SRC” and “SCR” mean the same thing? If that’s so, please use the same term throughout the paper.

The response (1): Many thanks for your valuable comments. Yes,  It is the same meaning, we have amended the wrong term in our manuscript.

2- The reviewer question:

2- p.3, Line 4 from the bottom. I don’t understand what “20mm” means. Does the 20mm have the same meaning as the maximum size of aggregate? Could you please clarify what you mean by this?

The response (2): Many thanks for your valuable comments. In recycling coarse aggregates (RCA) that had been used in the experiments maximum size of aggregate is 20 mm.

3- The reviewer question:

  1. p.3, Line 1 from the bottom

Please indicate the physical properties (density and water absorption) of waste tires. These values are important for confirming mix proportion of concrete.

The response (3): Many thanks for your appreciated comments

Your request has been done please  kindly check page 4 (Table 1:Coarse and fine Aggregates properties)

4- The reviewer question:

4-p.4, 2.2 Cement

Please explain why you used sulfate resistance cement. And please also explain the reason for adding silica fume and fly ash.

The response (4): Many thanks for your respected comments. Kindly check page (5) 

 5- The reviewer question:

  1. p.5, Line 7 from the bottom “binder content of 507 kg/m3”

In Table 5, cement weight is 396 kg/m3. Which value is correct?

 The response (5): cement weight is 396 kg/m3 was corrected

6- The reviewer Question  

  1. p.5, Line 6 from the bottom

“2% (by weight) replacement”

In Table 5, FA% of the cement content is 0.20%. Which value is correct?

The Response (6): FA% of the cement content is 0.20%  has been corrected

7. p.8, Line 8 from the bottom‎

7- The reviewer Question  Please explain the reason the slump increases with adding WRT.‎

 The response (7): The slump increases with adding SF and fly ash at WRT at [OPC]and [SRC] because ‎ water ‎absorption of WRT increases, The initial porosity of WRT increases compared to the ‎reference RF, and the density decreased. ‎

8- The reviewer question 8

8. p.9, Line 3 from the bottom‎

‎“The disadvantage of lowering air content may lead to reduce some of the desired effects of ‎tire rubber in concrete”‎

What does the desired effects mean? I don’t understand it.‎

The response(8): That means air-entrained concrete with low content of air voids. Optimal air entrainment ‎will protect concrete from the harmful effects of the interaction of sulfate attack and ‎freeze-thaw damage.‎

9- The reviewer question 9

P.12, 3.6 Ductility performance and Failure Mechanism

I think it is very difficult to evaluate ductility performance only from the result of crack failure behavior. Quantitative evaluation using indicators such as stress-strain curve and fracture energy is necessary.

The Response 9 : Thanks for your comment your request has been added please check page 13,14)

10- The reviewer Question 10

  1. p.13, Fig.9 Please add the name of the specimen (mix No.) in this photo. And the photo at the bottom is the result of tensile test, so please correct the sub-caption.

Furthermore, please add the meaning of the values in the bottom of photo.

Response 10: Your request has been done check page 14, 15 (fig 9)

  1. The reviewer question 11

‎ 11. p.14, 3.7 SEM .The discussion on the SEM images is very confusing.

Response 11:  kindly check fig 10 and  page 16-18

12.The reviewer question 12

12. p.18, 3.8.2 The relationship between porosity and the bulk density

I don’t understand what can be explained by this relationship. Please add in detail.‎

Response 12:  kindly check page 22

And  our manuscript has been sent for editing language check the (MDPI confirms that it has received payment of English editing invoiceenglish-44639 (invoice dated18 May 2022

We very much hope the revised manuscript is accepted for publication in your respected Recycling journal

Thanks

Lamiaa K Idriss

The corresponding author

Reviewer 4 Report

Despite the large amount of research and experimental data obtained, the article requires significant revision. There are many technical errors in the article: superfluous elements such as points, parentheses; breaks in sentences; capital letters in the middle of a sentence, which make it difficult to understand the article. There are also many abbreviations in the article that are not used correctly. It is recommended to make a serious proofreading of the material, check the correct spelling of abbreviations, their logical application. Please exclude the simultaneous use of the full term and abbreviation. This is allowed only at the first mention of the full term. In the PDF version of the article, the places requiring corrections are highlighted as notes.

Author Response

Dear  Professor (Reviewer)

Greetings

 We would like to thank you for the letter dated 13/5/2022, and for the opportunity to resubmit a revised copy of this manuscript.

We would also like to take this opportunity to express our thanks to the reviewers for the positive feedback and helpful comments regarding the correction or modification of our manuscript titled (Rubberized concrete properties with different cement types).
We believe have resulted in an improved revised manuscript, which you will find uploaded alongside this document.

And  our manuscript has been sent for editing language check the (MDPI confirms that it has received payment of English editing invoiceenglish-44639 (invoice dated18 May 2022, according to your request

We very much hope the revised manuscript is accepted for publication in your respected Recycling journal

Thanks

Lamiaa K Idriss

The corresponding author

Round 2

Reviewer 1 Report

This article is advised to be accepted.

Author Response

Dear  professor : (Reviewer:1)

Greetings

 We appreciate you for your precious time in re-reviewing the paper of our manuscript titled (Rubberized concrete properties with different cement types) on 22/5/2022and providing ‎valuable comments. It was your valuable and insightful comments that led to possible ‎improvements in the current version, we have carefully considered the comments.‎

 We very much thank you're advice for being accepted this article.

Lamiaa K Idriss

The corresponding author

Reviewer 2 Report

The authors have addressed all my comments. There are still some typos that need to be revised in the manuscript, for example, SRC or SCR is confusing.

Author Response

Dear  Professor : (Reviewer 2)

Greetings,

We appreciate you for your precious time in re-reviewing our manuscript titled (Rubberized concrete properties with different cement types) on 22/5/2022and providing ‎valuable comments. It was your valuable and insightful comments that led to possible ‎improvements in the current version. We hope the manuscript ‎after careful revisions meet your high standards.‎

. All modifications in the manuscript ‎

‎  ‎

1- The reviewer’s comment:

The authors have addressed all my comments. There are still some typos that need to be revised in the manuscript, for example, SRC or SCR is confusing.

The response:

The correction has been done, kindly check the manuscript after modification.

Thanks

Lamiaa K Idriss

The corresponding author

Reviewer 3 Report

The paper has been revised well. I think this paper will be acceptable after some corrections have been done.

1. “SCR” replace by “SRC” throughout the paper.

2. In the legend of Fig.11, the legend of “Poly.(Bulk Density of OPC)” is duplicated.

Author Response

Dear Professor: (Reviewer 3)

Greetings,

We appreciate you for your precious time in re-reviewing the paper of our manuscript titled (Rubberized concrete properties with different cement types) on 22/5/2022and providing ‎valuable comments. It was your valuable and insightful comments that led to possible ‎improvements in the current version, we have carefully considered the comments. We hope the manuscript ‎after careful revisions meet your high standards. ‎

Here is a point-by-point response to the reviewers’ comments and concerns.

 1- The reviewer’s comment:

  1. “SCR” replace with “SRC” throughout the paper.

The response:

The correction has been done, kindly check the manuscript after modification.

2- The reviewer’s comment:

  1. In the legend of Fig.11, the legend of “Poly.(Bulk Density of OPC)” is duplicated.

The response:

The correction has been done, kindly check Fig.11. after modification.

We again appreciate your kindness in your helping to improve the manuscript. Thanks

Lamiaa K Idriss

The corresponding author

Reviewer 4 Report

The authors partially corrected the technical comments. However, the manuscript still needs to be corrected. For example:

- Comment 2 is fixed partially; - For Comment 9: no adjustments were made -Comment 10. Microscope data should be provided in Section 2; - Comment for Response 19; - Please explain how "superplasticizer promotes the dispersion of cement particles and interacts with Ca(OH)2". This opinion is controversial. Therefore, it is recommended to add literature sources that confirm the "dispersive" effect of the superplasticizer and its interaction with Ca(OH)2. - There are still a lot of technical errors in the article. The numbering of the tables is broken (two tables numbered 3, and there is no table number 4). Authors are encouraged to proofread the text again and make other corrections.

– Table 3. Replace the sentence “The properties of Silica Fume [SF] and Fly ash from the manufacturer data sheet” with “The properties of SF and Fly ash from the manufacturer data sheet”.

Remove "[]" for all abbreviations that appear in the text of the article not for the first time

– Table 5, Figures 5, 11. Replace “Kg/m3” with “kg/m3” – For reference sources in articles, use "[]" instead of "()". In all other cases, use "()" in the text instead of "[]"

 – Tables 9, 11, 13–15. Use "0". For example, "0.6" instead of ".6".

Correct spelling is shown in Tables 5–7. Notes cited in articles as footnotes have not been corrected, as well.

Author Response

Dear Professor: (Reviewer 4)

Greetings,

We appreciate you for your precious time in re-reviewing the paper of our manuscript titled (Rubberized concrete properties with different cement types) on 22/5/2022and providing ‎valuable comments. It was your valuable and insightful comments that led to possible ‎improvements in the current version, we have carefully considered the comments and tried our best to address every one of them. We hope the manuscript ‎after careful revisions meet your high standards.‎

Here is a point-by-point response to the reviewers’ comments and concerns

1- The reviewer’s comment:

Comment 2 is fixed partially; [ When using abbreviations, it is recommended to replace [] with (), for example,[WT] replace with waste tires (WT)]

The response:

The correction has been done, kindly check the manuscript.

2- The reviewer’s comment:

 For Comment 9: no adjustments were made [It is recommended not to highlight the purpose of the study in a separate subsection 1.1].

The response:

The correction has been done, kindly check section 2 page 3.

3- The reviewer’s comment:

Comment 10. Microscope data should be provided in Section 2;[ Indicate the company and country of manufacture of Scanning Electron Microscope.           

The response:

Microscope data has been provided, kindly check the section2 page3.

4- The reviewer’s comment:

Comment for Response 19; - Please explain how "superplasticizer promotes the dispersion of cement particles and interacts with Ca(OH)2". This opinion is controversial. Therefore, it is recommended to add literature sources that confirm the "dispersive" effect of the superplasticizer and its interaction with Ca(OH)2.

The response:

explain how "superplasticizer promotes the dispersion of cement particles and interacts with Ca(OH)2" and recommended adding literature sources. [26], kindly check page 5, section2.2.

5- The reviewer’s question:

 - There are still a lot of technical errors in the article. The numbering of the tables is broken (two tables numbered 3, and there is no table number 4).

 Authors are encouraged to proofread the text again and make other corrections.

The response:

kindly check pages 5,6 tables numbered 3. a,3. b, and table 4.

6- The reviewer’s question:

– Table 3. Replace the sentence “The properties of Silica Fume [SF] and Fly ash from the manufacturer datasheet” with “The properties of SF and Fly ash from the manufacturer datasheet”.

The response:

kindly check page5 Tables 3. a and 3. b

7- The reviewer’s comment:

– Remove "[]" for all abbreviations that appear in the text of the article, not for the first time.

The response:

 We had Removed []" for all abbreviations that appear in the text of the article kindly check all the manuscript.

8- The reviewer’s comment:

– Table 5, Figures 5, 11. Replace “Kg/m3” with “kg/m3” –

The response:

We had Replaced, “Kg/m3” with “kg/m3, kindly check pages 6,8,20

9- The reviewer’s comment:

 For reference sources in articles, use "[]" instead of "()". In all other cases, use "()" in the text instead of "[]"

The response:

We had Replaced reference sources in articles, kindly check the manuscript after editing.

10- The reviewer’s comment:

 – Tables 9, 11, 13–15. Use "0". For example, "0.6" instead of ".6".

The response:

We had been corrected, kindly check page 17,8,19 for these tables.

 11- The reviewer’s comment:

Correct spelling is shown in Tables 5–7. Notes cited in articles as footnotes have not been corrected, as well.[ Table 5. Please remove %.]

The response:

We had been corrected, kindly check pages 6,9,10 for these tables.

We very much hope the revised manuscript is accepted for publication in your respected Recycling journal

Thanks

Lamiaa K Idriss

The corresponding author